# Local Linear Attention: An Optimal Interpolation of Linear and Softmax Attention For Test-Time Regression

**Yifei Zuo**
Northwestern University
Evanston, IL 60208, USA
`yifeizuo2029@u.northwestern.edu`

**Yutong Yin**
Northwestern University
Evanston, IL 60208, USA
`yutongyin2028@u.northwestern.edu`

**Zhicheng Zeng**
University of Washington
Seattle, WA 98195, USA
`zczeng@uw.edu`

**Ang Li**
University of Washington
Seattle, WA 98195, USA
`angliz@uw.edu`

**Banghua Zhu**
University of Washington
Seattle, WA 98195, USA
`banghua@uw.edu`

**Zhaoran Wang**
Northwestern University
Evanston, IL 60208, USA
`zhaoranwang@gmail.com`

## Abstract

Transformer architectures have achieved remarkable success in various domains. While efficient alternatives to Softmax Attention have been widely studied, the search for more expressive mechanisms grounded in theoretical insight—even at greater computational cost—has been relatively underexplored. In this work, we bridge this gap by proposing Local Linear Attention (LLA), a novel attention mechanism derived from nonparametric statistics through the lens of test-time regression. First, we show that LLA offers theoretical advantages over Linear and Softmax Attention for associative memory via a bias-variance trade-off analysis. Next, we address its computational challenges and propose two memory-efficient primitives to tackle the $\Theta(n^2d)$ and $\Theta(nd^2)$ complexity. We then introduce Flash-LLA, a hardware-efficient, blockwise algorithm that enables scalable and parallel computation on modern accelerators. In addition, we implement and profile a customized inference kernel that significantly reduces memory overheads. Finally, we empirically validate the advantages and limitations of LLA on test-time regression, in-context regression, associative recall and state tracking tasks. Experiment results demonstrate that LLA effectively adapts to non-stationarity, outperforming strong baselines in test-time training and in-context learning, and exhibiting promising evidence for its scalability and applicability in large-scale models.

## 1 Introduction

Transformer-based architectures have dominated modern AI systems and powered breakthroughs across various fields. The core computation primitive–Softmax Attention (Vaswani et al., 2017), adaptively processes and aggregates contextual information. Recent research on architecture innovation has proposed numerous variants of attention mechanism such as Linear Attention (LA) (Yang et al., 2024b; 2025; Siems et al., 2025; Sun et al., 2023; Ma et al., 2024; Liu et al., 2024; Katharopoulos et al., 2020; Yang et al., 2024a) and state space models (SSMs) (Gu et al., 2022a;b; Gu & Dao, 2024; Dao & Gu, 2024; Poli et al., 2023). While these methods offer remarkable efficiency improvements on long sequences, they often incur a performance penalty compared to Softmax Attention (Bick et al., 2025; Katharopoulos et al., 2020). Meanwhile, many of the design choices such as gating and forgetting factor (Yang et al., 2024a; 2025; Lin et al., 2025; Gers et al., 2000) are often

guided by heuristics or empirical results, lacking a principled understanding. Conversely, the search for more expressive attention mechanisms, even at an additional computational cost, has been relatively under-explored. Recently, test-time regression (Wang et al., 2025) unifies the design choices of different attention variants, indicating that the attention mechanism can be viewed as a test-time optimizer for layer-specific regression problems. A natural question arises: *can we systematically improve the Softmax Attention mechanism from the perspective of test-time regression?*

In this work, we propose *local linear attention (LLA)*, an upgrade to Softmax Attention derived from local linear regression. Our contributions are summarized as follows:

- We systematically analyze the design space of attention mechanisms within the test-time regression framework and propose LLA. We provide theoretical comparison of LLA with Softmax Attention and LA family from the perspectives of bias-variance trade-off and demonstrate its provable advantage in associative recall capability.

- We provide a detailed discussion on the computation of LLA and overcome the $\Theta(n^2 d)$ and $\Theta(nd^2)$ memory complexity with two optimizations, where $n$ is the sequence length and $d$ is the dimension. We then introduce FlashLLA, a blockwise and hardware-efficient algorithm to parallelize the computation on modern accelerators. In addition, we implement and profile a customized inference kernel with significant memory reduction.

- We conduct extensive experiments on synthetic tasks including test-time regression, in-context regression, associative recall and state tracking to validate the advantages and limitations of LLA compared to strong baselines.

## 1.1 RELATED WORKS.

**Linear Attention and State Space Models.** Due to the quadratic computational cost and linear memory consumption of the softmax attention mechanism for autoregressive sequence modeling, efficient attention mechanisms such as LA (Yang et al., 2024b; 2025; Siems et al., 2025; Sun et al., 2023; Ma et al., 2024; Liu et al., 2024; Yang et al., 2024a) and SSMs (Gu et al., 2022a; Gu & Dao, 2024; Dao & Gu, 2024; Poli et al., 2023) were proposed in search of more efficient alternatives for long-context sequence generation. These methods maintain a constant sized hidden state (Katharopoulos et al., 2020) during decoding and update it like a linear RNN. This state update behavior is also referred as fast-weight programming (Schmidhuber, 1992; Schlag et al., 2021). Essentially, MesaNet (von Oswald et al., 2025) is a LA variant that preconditions the hidden state and achieves optimal regression objectives among linear models.

**In-Context Learning by Optimization.** A growing body of work suggests that attention mechanism implicitly performs optimization algorithms to achieve in-context learning (Garg et al., 2022; Akyürek et al., 2023; von Oswald et al., 2023; Kirsch et al., 2024; Zhang et al., 2024; Mahankali et al., 2024; Ahn et al., 2023; Dai et al., 2023). For example, Mesa optimization (von Oswald et al., 2023) suggested that the attention layer inherently performs or approximates optimization steps during the forward pass. This behavior is particularly evident in LA and SSMs, as the hidden state update can be interpreted as performing gradient descent to solve a linear regression objective (Wang et al., 2025; Liu et al., 2024). MesaNet (von Oswald et al., 2025) is a one-step convergent algorithm for such a problem as the objective accepts a closed-form solution.

**Hardware Efficient Attention.** Prior works have focused on efficient attention implementation on modern hardwares to alleviate memory and computation overheads. FlashAttention (Dao et al., 2022; Dao, 2024) performs a block-wise online softmax to reduce I/O latency. Several other approaches, including NSA (Yuan et al., 2025), SeerAttention (Gao et al., 2024), MoBA (Lu et al., 2025), and Block Sparse Attention (Xiao, 2025), leverage sparsity to lower the effective computational cost while preserving GPU utilization. Flash Linear Attention (Yang et al., 2024a) provides a hardware-friendly formulation of linear attention through chunk-wise computation.

## 1.2 NOTATION.

We use upper-case letters to denote matrices and lower-case letters to denote vectors. For a matrix $X$, we denote its Frobenius norm as $\|X\|_F$ and the Hadamard product as $\odot$. For a vector $x$, we denote its

Euclidean norm as $\|x\|_2$. Furthermore, define $\mathtt{rsum}(X) = X\mathbf{1}$ for matrix $X$ and $\mathtt{bcast}(x) = x\mathbf{1}^\top$ for vector $x$, where $\mathbf{1}$ is a vector of ones. We use the abbreviation $\mathtt{brsum}(x) = \mathtt{bcast}(\mathtt{rsum}(x))$.

## 2 BEYOND LOCAL CONSTANT ESTIMATE

In this section, we first revisit the test-time regression interpretation for the attention mechanism (Wang et al., 2025). Then, we analyze the associative recall capacity and show the inherent limitations of LA and Softmax Attention. Lastly, we introduce the formulation for LLA.

### 2.1 ATTENTION AS TEST-TIME REGRESSION

In test-time regression framework, the attention mechanism is interpreted as a layer-specific regression solver. The goal is to approximate an unknown regression function $f : \mathbb{R}^d \mapsto \mathbb{R}^d$ using historical key-value pairs. To be specific, given a hypothesis space $\mathcal{F}$ and a position $1 \leq i \leq n$, an estimator $\hat{f}_i \in \mathcal{F}$ is fitted on the dataset $\mathcal{D}_i = \{(k_j, v_j) \in \mathbb{R}^d \times \mathbb{R}^d\}_{j=1}^i$, where the attention keys $k_j$ serve as the features and attention values $v_j$ as the labels. The prediction is made at a query $q_i \in \mathbb{R}^d$, which is treated as a test data point.

**Linear Attention as Parametric Regression.** Parametric model constrains the function class $\mathcal{F}$ to a set of functions defined by a finite-dimensional parameter $\theta \in \Theta$. The most fundamental instantiation is the linear regression, which sets $\mathcal{F} = \{f_\theta(x) = Wx + b \mid \theta = (W, b), W \in \mathbb{R}^{d \times d}, b \in \mathbb{R}^d\}$. We omit the intercept $b$ for simplicity. At each position $i$, the parameter $W_i$ is estimated by solving the following least square problem on the training dataset $\mathcal{D}_i$,

$$\min_W \mathcal{L}(W; \mathcal{D}_i) = \frac{1}{2} \sum_{j=1}^i \gamma_{ij} \|v_j - Wk_j\|_2^2 + \lambda \|W\|_F^2, \tag{1}$$

where $\gamma_{ij} \in \mathbb{R}$ is a weighting factor and $\lambda \geq 0$ is the ridge regularization penalty. For appropriate regularization, objective equation 1 admits a closed-form optimal solution. MesaNet (von Oswald et al., 2025; 2024) hardcodes this solution for the case $\gamma_{ij} = 1$, where the prediction is given by,

$$\hat{f}_{\mathtt{Mesa}}(q_i) = \hat{W}_i^{\mathtt{Mesa}} q_i = \underbrace{\left(\sum_{j=1}^i v_j k_j^\top\right)}_{S_i} \underbrace{\left(\sum_{j=1}^i k_j k_j^\top + \lambda I\right)^{-1}}_{H_i} q_i. \tag{2}$$

Across different position $i$, the weight $W_i^{\mathtt{Mesa}}$ can be updated recurrently by maintaining two statistics $S_i$ and $H_i$ with $\Theta(d^2)$ memory. This allows MesaNet to be interpreted as a linear RNN with two recurrent states. To avoid the expensive matrix inversion, vanilla LA brutally approximates the precondition matrix $H_i \approx I$ in equation 2, leading to a suboptimal solution of the least square problem. It can also be shown that LA variants and SSMs such as GLA (Yang et al., 2024a), RetNet (Sun et al., 2023), RWKV (Peng et al., 2024; 2025) and Mamba (Gu & Dao, 2024; Dao & Gu, 2024) can be derived by the approximation $H_i \approx I$ with different weighting schemes $\gamma_{ij}$. Besides exact solutions, vanilla LA and variants such as DeltaNet (Yang et al., 2024b) and Gated DeltaNet (Yang et al., 2025) can be interpreted as performing one step of first order stochastic gradient descent on weight $W$. We refer readers to (Wang et al., 2025) and tables in (Peng et al., 2025; Yang et al., 2025) for detailed derivations and how each model is implemented.

**Softmax Attention is Non-Parametric.** Non-parametric regression makes minimal structural assumptions on the function class $\mathcal{F}$. A canonical example is the kernel regression, where the estimator $\hat{f}$ is defined directly from the data in a way that depends on the query point. Specifically, for a query vector $q_i$, let $\mathcal{F}(q_i)$ denote the local function class around $q_i$. The local regression objective is defined as:

$$\min_{f \in \mathcal{F}(q_i)} \mathcal{L}(f; \mathcal{D}_i) = \frac{1}{2} \sum_{j=1}^i w_{ij} \|v_j - f(k_j)\|_2^2, \tag{3}$$

where $w_{ij} = K_h(q_i, k_j) \in \mathbb{R}$ is a query-dependent weight that measures the locality of the training point $k_j$ to the query $q_i$. The simplest instantiation is the constant model, where $\mathcal{F}(q_i) = \{f_\theta(x) = \theta \in \mathbb{R}^d, \forall x \in \mathbb{R}^d\}$. Solving objective equation 3 with this function class yields:

$$\hat{f}(q_i) = \sum_{j=1}^{i} s_{ij} v_j, \quad s_{ij} = \frac{w_{ij}}{\sum_{j'=1}^{i} w_{ij'}} \tag{4}$$

Consider an RBF kernel $w_{ij} = \exp(-\|k_j - q_i\|^2/h)$ with bandwidth $h = 2\sqrt{d}$, the estimator equation 4 exactly recovers the softmax attention when QK normalization (Dehghani et al., 2023; Wortsman et al., 2023; Team, 2024) is applied, since in this case $w_{ij} \propto \exp(q_i^\top k_j/\sqrt{d})$ and the common constant factor cancels in the division. It is also known as the Nadaraya-Watson (NW) kernel regression (Nadaraya, 1964; Watson, 1964; Bierens, 1988) in statistics literature. We note that QK normalization is not strictly necessary to represent practical Softmax Attention as the additional term can naturally serve as positional encoding (Press et al., 2022), and the scale effectively tunes the bandwidth $h$ in a data-dependent manner.

## 2.2 Learning Behavior of Associative Recall

Attention mechanisms are often evaluated by the associative memory capacity (Zhong et al., 2025; Behrouz et al., 2026; Ramsauer et al., 2021). Specifically, given a training set of key-value pairs $\{(k_j, v_j)\}_{j=1}^{i}$, the model is expected to retrieve the value $v_j$ associated with $k_j$ when queried at $q = k_j$. This objective can be exactly captured by the Mean Square Error (MSE). For example, the retrieval error in vanilla LA is given by,

$$\text{MSE}_i^{\text{LA}} = \frac{1}{i} \sum_{j=1}^{i} \|S_i k_j - v_j\|^2 = \frac{1}{i} \sum_{j=1}^{i} \|(k_j^\top k_j - 1)v_j + \sum_{j' \neq j} v_{j'} k_{j'}^\top k_j\|^2, \tag{5}$$

The first term in the summation is the signal bias and can be avoid by QK normalization. The second term is the interference from other key-value pairs. Deltaformer (Zhong et al., 2025) quantitatively analyze this error by the inverse of signal-to-noise ratio $\text{SNR}^{-1}$, which is essentially a normalized version of MSE. In classic literature, MSE decomposition allows us to analyze the approximation and generalization error of the model and the corresponding bias-variance trade-off.

**Irreducible Approximation Error of Global Linear Model.** Recall that global linear models correspond to solving a least square problem equation 1 over the hypothesis space $\mathcal{F} = \{f_\theta(x) = Wx + b\}$. When the ground truth function $f$ is not global linear, any estimator $\hat{f} \in \mathcal{F}$ will suffer from a non-vanishing approximation error due to model misspecification. In contrast, the local constant model is a nonparametric estimator that does not impose structural assumptions on the function class except for smoothness or regularity. Consequently, the approximation error vanishes asymptotically with proper assumptions. In fact, we have the following separation result between global linear (GL) and local constant (NW) estimators:

**Proposition 2.1.** *Let* $(X_i, Y_i)_{i=1}^{n}$ *be i.i.d.,* $X_i \in \mathbb{R}^d$ *supported on a bounded set* $D \subset \mathbb{R}^d$*, and* $Y_i = f(X_i) + \varepsilon_i \in \mathbb{R}^{d_y}$ *with* $\mathbb{E}[\varepsilon_i \mid X_i] = 0$ *and* $\mathbb{E}[\varepsilon_i^2 \mid X_i] = \sigma^2(X_i)$*. Let* $\hat{f}_{\text{GL}}$ *denote a global-linear estimator and* $\hat{f}_{\text{NW}}$ *the local-constant (NW) estimator with optimal bandwidth. Under mild assumptions, if* $f$ *is not globally linear, then*

$$\mathbb{E} \int_D \|\hat{f}_{\text{GL}}(x) - f(x)\|^2 dx = \Omega(1) , \ \mathbb{E} \int_D \|\hat{f}_{\text{NW}}(x) - f(x)\|^2 dx = O(n^{-3/(d+3)}).$$

**Irreducible Boundary Bias of Local Constant Model.** Despite the appealing convergent property of local constant model, it suffers when predicting near the boundary of the data support, particularly with symmetric kernels like RBF. This phenomenon becomes more pronounced in high-dimension and likely to occur more frequently in autoregressive prediction. Local polynomial regression is a standard remedy in nonparametric statistics to address this issue. In Section 2.3, we will introduce LLA as a natural adaptation of local linear regression. In fact, we have the following separation result between local constant (NW) and local linear (LL) estimators:

**Proposition 2.2.** *Under the setting of Proposition 2.1, let $\widehat{f}_{\mathrm{NW}}$ and $\widehat{f}_{\mathrm{LL}}$ denote local-constant and local-linear estimators with their respective optimal bandwidths. Under mild assumptions, if $f$ has sufficiently large normal gradient along the boundary of D, then*

$$\mathbb{E}\int_D ||\widehat{f}_{\mathrm{NW}}(x) - f(x)||^2 dx = \Omega(n^{-3/(d+3)}) \,, \quad \mathbb{E}\int_D ||\widehat{f}_{\mathrm{LL}}(x) - f(x)||^2 dx = O(n^{-4/(d+4)}).$$

The proof of proposition 2.1 and 2.2 are provided in Appendix A and B respectively.

### 2.3 LOCAL LINEAR ATTENTION

**Formulation.** For a query $q_i$, instantiate the local function class $\mathcal{F}(q_i) = \{f_\theta(x) = b + W(x - q_i) \mid \theta = (W, b), W \in \mathbb{R}^{d \times d}, b \in \mathbb{R}^d\}$. The regularized local linear regression objective is,

$$\min_{f \in \mathcal{F}(q_i)} \mathcal{L}(f; \mathcal{D}_i) = \frac{1}{2}\sum_{j=1}^{i} w_{ij}\|v_j - b - W(k_j - q_i)\|^2 + \lambda\|W\|_F^2, \tag{6}$$

where $w_{ij}$ are query-dependent kernel weights and $\lambda \geq 0$ is a ridge penalty. This objective also admits a closed-form solution for the intercept $b$ and weight $W$. Importantly, at test time, the prediction is only made at $\hat{f}(q_i) = \hat{b}_i$. Thus it suffices to derive the formulation for the intercept. Define $z_{ij} = k_j - q_i$ and the following query-specific statistics,

$$\omega_i = \sum_{j=1}^{i} w_{ij} \in \mathbb{R}, \quad \mu_i = \sum_{j=1}^{i} w_{ij}z_{ij} \in \mathbb{R}^d, \quad \Sigma_i = \sum_{j=1}^{i} w_{ij}z_{ij}z_{ij}^\top + \lambda I \in \mathbb{R}^{d \times d}. \tag{7}$$

Also denote $\rho_i = \Sigma_i^{-1}\mu_i \in \mathbb{R}^d$. The optimal intercept can be computed as follows,

$$\hat{f}(q_i) = \hat{b}_i = \sum_{j=1}^{i} s_{ij}v_j, \quad s_{ij} = w_{ij}\frac{1 - z_{ij}^\top\rho_i}{\omega_i - \mu_i^\top\rho_i}. \tag{8}$$

Similar to maintaining $H_i$ in MesaNet, LLA requires capturing the precondition matrix $\Sigma_i$. However, a key difference is that $H_i$ is built on global statistics that are independent of the query, whereas $\Sigma_i$ is constructed from features centered around the specific query $q_i$ for each position $1 \leq i \leq n$. Consequently, LLA requires a KV cache of size $\Theta(nd)$ similar to Softmax Attention, rather than constant-size recurrent states as in the LA family.

**LLA Interpolates Linear and Softmax Attention.** A more interpretable form of equation 8 can be obtained by decomposing the prediction into two components. Suppose that the weight matrix $\hat{W}_i$ is given prior to solving equation 6, then the optimal intercept can be expressed as,

$$\hat{b}_i = \sum_{j=1}^{i} s_{ij}(v_j - \hat{W}_i k_j) + \hat{W}_i q_i, \quad s_{ij} = \frac{w_{ij}}{\sum_{j'=1}^{i} w_{ij'}}. \tag{9}$$

The first term is a local constant regression to predict the residuals $v_j - \hat{W}_i k_j$, while the second term is a linear prediction based on $\hat{W}_i$. The formulation recovers LLA if $\hat{W}_i$ is obtained by optimally solving equation 6. However, by allowing suboptimal estimation, one can construct $\hat{W}_i$ as a recurrent state similar to LA. This decomposition reveals how LLA interpolates between Linear and Softmax Attention and provides a template for designing new algorithms.

### 3 PRACTICAL ALGORITHM

In this section, we provide a detailed discussion of the computation involved in LLA. We highlight two major challenges in naïve implementations and develop a practical block-wise algorithm FlashLLA that scales efficiently on modern accelerators such as GPUs.

### 3.1 MEMORY EFFICIENT PRIMITIVES

**Avoid Pairwise Materialization.** The first bottleneck is the evaluation of vectors $z_{ij} = k_j - q_i$ for every $1 \leq j \leq i \leq n$, which requires $\Theta(n^2 d)$ memory to materialize. This pairwise difference is later used in the formulation of $\mu_i$ and $\Sigma_i$ defined in equation 7 as well as the inner product $z_{ij}^\top \rho_i$ in equation 8. For both cases, explicit materialization of $z_{ij}$ can be avoid by algebraically separating the contributions of $k_j$ and $q_i$ to the final result. Specifically, the statistics $\mu_i$ and $\Sigma_i$ can be reformulated in terms of intermediate quantities that are independent of the query and can then be transformed to recover the original centered statistics:

$$\tilde{\mu}_i = \sum_{j=1}^i w_{ij} k_j \in \mathbb{R}^d, \quad \tilde{\Sigma}_i = \sum_{j=1}^i w_{ij} k_j k_j^\top + \lambda I \in \mathbb{R}^{d \times d} \tag{10}$$

$$\mu_i = \tilde{\mu}_i - \omega_i q_i, \quad \Sigma_i = \tilde{\Sigma}_i - \tilde{\mu}_i q_i^\top - q_i \tilde{\mu}_i^\top + \omega_i q_i q_i^\top \tag{11}$$

The computation in equation 10 and equation 11 only requires vectors $k_j$ and $q_i$ individually, reducing the memory cost to $\Theta(nd)$. The same principle applies to computing inner products of the form $z_{ij}^\top x_i = k_j^\top x_i - q_i x_i$ for any vector $x_i \in \mathbb{R}^d$. The matrix operator **relative matrix multiplication (relmm)** for this optimization as follows,

$$\mathtt{relmm}(X, Q, K) := XK^\top - \mathtt{brsum}(X \odot Q). \tag{12}$$

This operator is invoked once in the forward computation with $x_i = \rho_i$, but appears multiple times in the backward with other variables. Further details on the backward are provided in Appendix D.

**Matrix-Free Inversion via Conjugate Gradients.** The second bottleneck arises in solving linear systems of the form $\Sigma_i^{-1} x_i$ for some vector $x_i \in \mathbb{R}^d$. Directly inverting $\Sigma_i$ for every $1 \leq i \leq n$ incurs a prohibitive $\Theta(nd^2)$ memory. Following the approach in MesaNet (von Oswald et al., 2025), we exploit the sum-of-rank-one structure of $\Sigma_i$ and solve the linear system iteratively using the conjugate gradient (CG) method (Hestenes & Stiefel, 1952). The key insight is that CG only evaluate the matrix-vector product $\Sigma_i p$ for a search direction $p \in \mathbb{R}^d$ without explicit matrix materialization:

$$\Sigma_i p = \sum_{j=1}^i w_{ij}(k_j^\top p)k_j - (q_i^\top p)\tilde{\mu}_i - (\tilde{\mu}_i^\top p)q_i + (\omega_i q_i^\top p)q_i + \lambda p. \tag{13}$$

Each term only involves inner products or weighted sums over keys and the query, both of which can be computed efficiently using batched matrix multiplication with $\Theta(nd)$ memory. This CG operation of $\Sigma_i$ is invoked once in the forward computation with $x_i = \mu_i$ and twice in the backward computation with other variables. Further details on the CG algorithm are provided in C.

### 3.2 PARALLEL FORM AND BLOCKWISE ALGORITHM

**Matrix Formulation.** We first express the key components of the LLA forward pass in matrix form. Let $Q, K, V \in \mathbb{R}^{n \times d}$ be the query, key, and value matrices respectively for a given layer and head. This function applies a causal mask to the input tensor using the $\mathtt{tril}$ operator that preserves the lower-triangular matrix. Then the output $O \in \mathbb{R}^{n \times d}$ can be computed as follows,

$$W = \mathtt{tril}(\exp(QK^\top/h)), \quad M = WK - \mathtt{brsum}(W) \odot Q \tag{14}$$

$$R = \mathtt{CGSolve}(M, Q, K, \lambda), \quad \delta = \mathtt{rsum}(W) - \mathtt{rsum}(M \odot R) \tag{15}$$

$$O = \left( \frac{1 - \mathtt{relmm}(R, Q, K)}{\mathtt{bcast}(\delta)} \odot W \right) V, \tag{16}$$

where $W \in \mathbb{R}^{n \times n}$ is the matrix of kernel weight $w_{ij}$, $M \in \mathbb{R}^{n \times d}$ stores the first-order statistics $\mu_i$, $R \in \mathbb{R}^{n \times d}$ contains the solution to the linear systems $\Sigma_i^{-1} \mu_i$ for every $1 \leq i \leq n$, and $\mathtt{CGSolve}(\cdot)$ invokes the CG algorithm that construct $\Sigma_i$ implicitly and solve these systems in parallel. The division and subtraction are performed element-wise. The single-head computation can be naturally extended to multi-head the same way as in standard multi-head attention mechanisms.

**Blockwise Algorithm.** Denote $B_r, B_c$ as the block size for queries and keys/values along the sequence length dimension and $r, c$ as the block index. Denote $Q_r \in \mathbb{R}^{B_r \times d}$ and $K_c, V_c \in \mathbb{R}^{B_c \times d}$ as the block-wise representations. The forward pass of FlashLLA is summarized in Algorithm 1.

---

**Algorithm 1** FlashLLA Forward Pass

---

**Require:** Matrices $Q, K, V$ in HBM, block sizes $B_r, B_c$, regularization $\lambda$, bandwidth $h$.
1: Divide $Q$ into $\lceil n/B_r \rceil$ blocks of size $B_r$ and $K, V$ into $\lceil n/B_c \rceil$ blocks of size $B_c$.
2: Divide output $O, R$ into $\lceil n/B_r \rceil$ blocks of size $B_r$.
3: **for** $r = 1$ to $\lceil n/B_r \rceil$ **do**
4:      Load $Q_r$ from HBM to SRAM.
5:      Initialize on-chip: $M_r^{(0)} \leftarrow 0 \in \mathbb{R}^{B_r \times d}, \omega_r^{(0)} \leftarrow 0 \in \mathbb{R}^{B_r}, m_r^{(0)} \leftarrow -\infty \in \mathbb{R}^{B_r}$
6:      **for** $c = 1$ to $\lceil n/B_c \rceil$ **do**
7:          Load $K_c$ from HBM to SRAM.
8:          Compute $W = Q_r K_c^\top / h$ and $m = \max(m_r^{(c-1)}, \texttt{rowmax}(W))$.
9:          Compute $\alpha_r = \exp(m_r^{(c-1)} - m), W = \exp(W - \texttt{bcast}(m))$ and update $m_r^{(c)} = m$.
10:         Compute $\omega_r^{(c)} = \alpha_r^{(c)} \odot \omega_r^{(c-1)} + \texttt{rsum}(W)$
11:         Compute $M_r^{(c)} = \texttt{bcast}(\alpha_r^{(c)}) \odot M_r^{(c-1)} + W K_c$.
12:      **end for**
13:      Initialize on-chip: $O_r^{(0)} \leftarrow 0 \in \mathbb{R}^{B_r \times d}, R_r^{(0)} \leftarrow 0 \in \mathbb{R}^{B_r \times d}$
14:      Compute $M_r = M_r^{(\texttt{last})} - \texttt{brsum}(W) \odot Q_r$.
15:      Compute $R_r = \texttt{CGSolve}(M_r, Q_r, K, M_r^{(\texttt{last})}, \omega_r^{(\texttt{last})}, \lambda)$.
16:      Compute $\delta_r = \omega_r^{(\texttt{last})} - \texttt{rsum}(M_r \odot R_r)$.
17:      **for** $c = 1$ to $\lceil n/B_c \rceil$ **do**
18:         Load $K_c, V_c$ from HBM to SRAM.
19:         Compute $W = \exp(Q_r K_c^\top / h - \texttt{bcast}(m_r^{(\texttt{last})}))$.
20:         Compute $S = (1 - \texttt{relmm}(R_r, Q_r, K_c)) \odot W/\texttt{bcast}(\delta_r)$.
21:         Compute $O_r^{(c)} = O_r^{(c-1)} + S V_c$.
22:      **end for**
23: **end for**

---

Since the statistics equation 7 for each query are computed independently, the forward pass of LLA can be naturally made parallel for batched queries. Therefore, the algorithm proceeds by iterating over query blocks $r$. Within each query block, the iteration over key/value blocks has three passes. (i) The first pass (line 6-12) corresponds to accumulating the statistics $M_r, \omega_r$ in an online fashion. Similar to online softmax (Ye, 2023; Milakov & Gimelshein, 2018), we maintain a running maximum $m_r$ to ensure numerical stability when computing the kernel weights. This trick is valid as the computation equation 8 is homogeneous in $w_{ij}$. (ii) The second pass (line 15) is encapsulated in the $\texttt{CGSolve}(\cdot)$ operator (see Appendix C for details). (iii) The third pass (line 17-22) computes the final output $O_r$ using the pre-computed results and the values $V_c$. To save computation in backward pass, we also store the intermediate $R_r$ and denominator $\delta_r$ alongside the output into HBM.

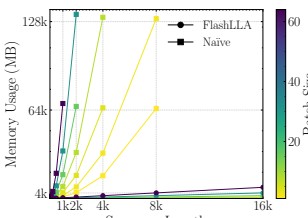

Figure 1: FlashLLA reduces working set memory to $\Theta(nd)$. The figure shows the profiling result for $d = 128$, OOM points are omitted.

We implement and benchmark the algorithm 1 in a custom $\texttt{Triton}$ kernel (∼500 lines of Python) across a range of dimensions and batch sizes on a single NVIDIA H200 GPU. Figure 1 demonstrate the quadratic dependency and quickly runs out of memory for naïve method. In contrast, the block-wise $\texttt{Triton}$ kernel significantly reduces the working set and scales linearly with sequence length, making it hardware-efficient and feasible for long-context and large batch training or inference. We also provide additional analysis and latency profiling in Appendix E.

## 4 EMPIRICAL RESULTS

**Test-Time Regression on Non-Stationary Data.** We first devise a synthetic piecewise-linear regression task to isolate the test-time adaptation capabilities of different attention mechanisms directly without training the query, key and value projections. Each sample is a length-$L$ sequence partitioned into $L/S$ contiguous segments with $S$ being the segment size. For each $c \in \{1, \dots, L/S\}$,

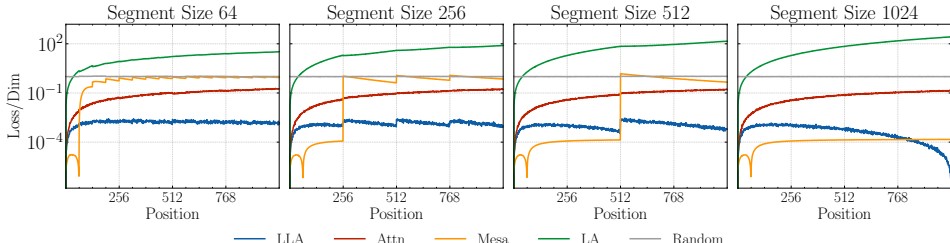

Figure 2: Test-time regression performance on a piecewise-linear task. The figures demonstrate position-wise MSE for $d = 64$ with $S \in \{64, 256, 512, 1024\}$. Results are averaged over 10,000 independently sampled sequences; LLA outperforms other baselines and benefits from more in-segment data; MesaNet excels only before the first shift. The y-axis uses a logarithmic scale.

keys $k_i \in \mathbb{R}^d$ are drawn from a segment-specific distribution $P_c$ supported on a distinct cone in the input space (see Appendix F for construction details). The corresponding values $v_i$ are generated by a segment-specific linear function $v_i = A_c k_i + \epsilon_i$ for $i \in \{(c-1)S+1, \dots, cS\}$ where $A_c \sim \mathcal{N}(0, I)$ and $\epsilon_i \sim \mathcal{N}(0, \delta^2 I)$. This design ensures the generated data $\{(k_i, v_i)\}_{i=1}^{L}$ has non-stationary input distribution and conditional mapping $f_c(k) = A_c k$.

We evaluate a single layer of each candidate model, including LLA, Softmax Attention, vanilla LA (Katharopoulos et al., 2020), MesaNet (von Oswald et al., 2025) as well as a random predictor. As this is a test-time only evaluation, we exclude mechanisms that require training to adapt. We set $L = 1024$ and sweep over different segment sizes $S$ and input dimensions $d$. Performance is measured by the position-wise MSE $\ell_i = \|\hat{f}(k_i) - v_i\|_2^2$ for $i \in \{1, \dots, L\}$ to capture the adaptation capability along the sequence. We also investigate the scaling behavior by evaluating the MSE ratio $\sum_{j=1}^{L} \ell_j^{\texttt{Model}} / \sum_{j=1}^{L} \ell_j^{\texttt{LLA}}$ for each model compared to LLA across different values of $d$ and $S$. The results are summarized in Figure 2. MesaNet achieves the best performance in the first segment where all the data is drawn from one linear mapping, as it effectively learns the optimal global linear function. However, it has limited capacity to adapt to distribution shifts and its performance degrades quickly in subsequent segments, where the data distribution does not match the learned global mapping.

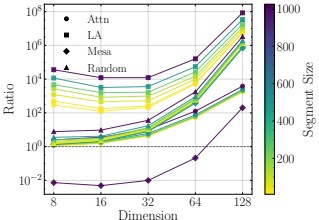

Figure 3: The advantage of LLA scales with the data dimension $d$. Both axes use a logarithmic scale.

Meanwhile, LLA continues to improve within each segment whereas Softmax Attention does not benefit from more in-distribution data. Moreover, the advantage of LLA scales favorably with data dimensionality (Figure 3), indicating the potential for adaptation to larger models and datasets.

**In-Context Regression on Non-Stationary Data.** We next evaluate the models' ability to perform in-context regression on non-stationary, piecewise-linear data. The data generation process follows the same principle as in the test-time regression task. The data points $\{x_i \in \mathbb{R}^{d_x}\}$ are generated from segment-specific distributions $P_c$ and the target is given by $y_i = A_c x_i + \epsilon_i \in \mathbb{R}^{d_y}$ for $i \in \{(c-1)S+1, \dots, cS\}$, where $A_c \sim \mathcal{N}(0, I_{d_y \times d_x}/d_x)$ and $\epsilon_i \sim \mathcal{N}(0, \delta^2 I_{d_y})$. Query $x' \in \mathbb{R}^{d_x}$ is randomly sampled from the segment distributions $P_c$. Each in-context regression prompt $X$ is constructed by concatenating $L$ shuffled input-target pairs with $L'$ queries:

$$X = \begin{pmatrix} x_1 & x_2 & \cdots & x_L & x'_1 & \cdots & x'_{L'} \\ y_1 & y_2 & \cdots & y_L & 0 & \cdots & 0 \end{pmatrix} \in \mathbb{R}^{(d_x + d_y) \times (L + L')}. \quad (17)$$

In contrast to the test-time regression setting, the query, key and value projections are parameterized. The model $f_\theta : \mathbb{R}^{d_x + d_y} \to \mathbb{R}^{d_y}$ is trained to predict the target $Y$, where the label to each query is generated by $y'_i = A_c x'_i$. Specifically, for a dataset $\mathcal{D}_{\texttt{train}} = \{(X^{(b)}, Y^{(b)})\}_{b=1}^{B}$, we minimize the MSE loss $\mathcal{L}(\theta; \mathcal{D}_{\texttt{train}})$ on the query tokens and report the test error $\mathcal{L}(\theta^*; \mathcal{D}_{\texttt{test}})$ after training.

We compare LLA against several strong baselines, including Softmax Attention, Mamba (Gu & Dao, 2024; Dao & Gu, 2024), GLA (Yang et al., 2024a), Hyena (Poli et al., 2023) and Gated DeltaNet

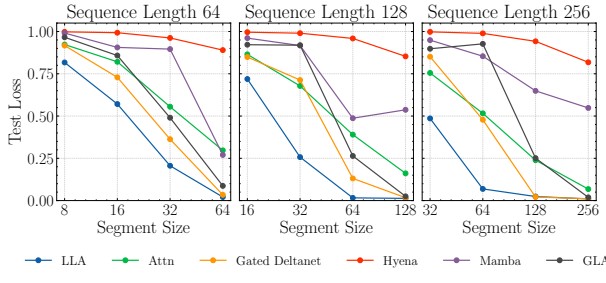 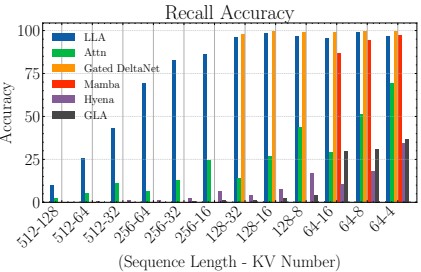

(a) Test Error of In-Context Regression.

(b) Test Accuracy of Associative Recall.

Figure 4: Figure a and b shown for models with $d = 128$ and 2 attention heads. Each point represents the best performance achieved across training hyperparameters, averaged over 3 random seeds.

(Yang et al., 2024b; 2025). We fix the dimension $d_x = d_y = 32$ and query number $L' = 16$ for all sweeps over segment sizes and evaluate two-layer models without MLPs. The results in Figure 4a demonstrate similar trends as in the test-time regression task, where LLA consistently outperforms other baselines across all configurations, particularly with smaller segment sizes.

**In-Context Associative Recall.** In-context recall is a fundamental capability of language models which requires the model to retrieve relevant information from the context based on the query. We adopt the MQAR task in Zoology (Arora et al., 2023) to evaluate this ability. Specifically, given two alphabets $\mathcal{A}_k, \mathcal{A}_v$ and a set of key-value pairs $(k_i, v_i) \in \mathcal{A}_k \times \mathcal{A}_v$, the set $\{k_i \mapsto v_i\}$ defines a many-to-one key-value association. The model is prompted with a sequence of key-value pairs and then queried with keys sampled from the context to predict the corresponding value.

As indicated in (Wang et al., 2025), a single short convolution layer is sufficient to solve next-token recall tasks. Therefore, we disable short convolution in all baselines to ensure fair comparison. We set $\|\mathcal{A}_k \cup \mathcal{A}_v\| = 8k$ and sweep over different sequence lengths and number of KV pairs. The test recall accuracy results in Figure 4b indicate that the advantages of LLA can be effectively transferred to discrete token prediction tasks. Additionally, we also observe different learning dynamics between LLA and Gated DeltaNet in this task. The results are discussed in Appendix F.4.

**Permutation State-Tracking.** We then test models' state-tracking ability by permutation state-tracking task. Given an initial assignment of items to positions, a sequence of swap instructions, and query positions. The model is trained to predict the item at each query position after all swaps. Each example is constructed as

$$\underbrace{p_1 = a_1, \, p_2 = a_2, \, \ldots, \, p_N = a_N}_{\text{initial state}} \; \# \; \underbrace{i_1 \, j_1, \, i_2 \, j_2, \, \ldots, \, i_S \, j_S}_{\text{swap instruction}} \; \# \; \underbrace{q_1 = o_1, \, \ldots q_{N'} = o_{N'}}_{\text{query + answer}}.$$

Here $p_n \in \{1, \ldots, N\}$ denotes position $n$; $a_n \in \mathcal{A}$ is the item initially assigned to $n$; each $(i_s, j_s) \in \{1, \ldots, N\}^2$ is a swap instruction exchanging the items at positions $i_s$ and $j_s$; and $q$ is the queried position. The target is the item at position $q$ after applying all $S$ swaps. We include explicit delimiter tokens #, =, and , for structure.

We draw the number of swaps as $S \sim \text{Uniform}(N/6, N/3)$ and set $|\mathcal{A}| = 8k$ for each example. The results in Figure 5 show that LLA achieves test accuracy on par with Softmax Attention across $N$. This outcome is expected from a complexity-theoretic perspective: constant-depth Softmax Attention is no more expressive than constant-depth threshold circuits $\mathsf{TC}^0$ and has limited ability to realize unbounded-depth state-tracking as $N$ grows (Hahn, 2020; Merrill & Sabharwal, 2023). By Eqs. 7 and 8, LLA augments Softmax

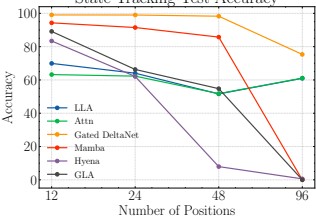

Figure 5: Test Accuracy of State Tracking. Results show the best score averaged over 3 random seeds.

Attention with a query-specific first-order correction computed via a constant number of parallel algebraic passes (weighted sum, inner product, inverse), which adds at most a constant extra circuit

layer. Its performance therefore matches the theoretical limits of Softmax Attention, explaining the results in Figure 5.

## 5 LIMITATIONS AND FUTURE DIRECTIONS

**High Computation and I/O Intensity.** Despite the significant reduction in memory consumption, LLA's computational cost remains substantially higher than that of Softmax Attention, primarily due to the matrix inversion involved in the computation. Exploring approximations to reduce the computation is an important direction for future work. Furthermore, while FlashLLA achieves the same $\Theta(nd + n^2)$ I/O complexity as in FlashAttention when the number of CG iterations is set as a constant (which is sufficient in practice), the constant factor is still higher due to the additional reads and writes required by the iterative solver. Incorporating hardware-aware optimizations, such as sliding windows or sparsity, could further reduce I/O complexity.

**Kernel Development and Evaluation on LLMs.** This work evaluates LLA on synthetic and moderate-scale tasks; its efficacy on large language models remains an ongoing question. Training LLMs with LLA using `PyTorch` implementation is infeasible due to its high computational and memory complexity. Therefore significant engineering efforts are required to stabilize and optimize the forward and backward kernel. Additionally, the numerical sensitivity of the matrix inversion poses a challenge for developing low precision kernels without sacrificing performance.

**Efficient Interpolation of Linear and Softmax Attention.** As shown in equation 9, LLA provides an optimal interpolation between Linear and Softmax Attention in solving the regression objective. And the formulation also provides a template to design algorithm for better computational efficiency while still retain strong estimation capabilities and potentially even improve upon the circuit complexity of Softmax Attention. For instance, future work could explore the integration of state-of-the-art Linear Attention architectures such as DeltaNet and Mamba using this template.

## REPRODUCIBILITY STATEMENT

The proofs of theoretical results 2.1 and 2.2 are provided in Appendix A and B, respectively. We also provide additional analysis and discussion on the implementation and profiling results in Appendix E. In Appendix F, we provide detailed instructions, configurations and additional experimental results for reproducing all the experiments in Section 4.

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

## A APPENDIX: PROOF OF PROPOSITION 2.1

Let $(X_i, Y_i)_{i=1}^n$ be i.i.d. with $X_i \in \mathbb{R}^d$ supported on a bounded domain $D \subset \mathbb{R}^d$ with density $p$, and

$$Y_i = f(X_i) + \varepsilon_i \in \mathbb{R}^{d_y}, \qquad \mathbb{E}[\varepsilon_i \mid X_i] = 0, \ \ \mathbb{E}[\varepsilon_i^2 \mid X_i] = \sigma^2(X_i).$$

The global linear estimator at $x \in D$ is

$$\widehat{f}_{\mathrm{GL}}(x) = \widehat{\theta}^\top (1, x^\top)^\top,$$

where $\widehat{\theta} \in \mathbb{R}^{d+1}$ minimizes the empirical squared loss over the global affine class.

Let $K : \mathbb{R}^d \to [0, \infty)$ be a bounded, compactly supported, radially symmetric kernel with $\int K(u) \, du = 1$. For a symmetric p.d. bandwidth matrix $H = H_n \succ 0$ with a constant condition-number upper bound $\kappa_1$, write

$$K_H(u) := |H|^{-1/2} K(H^{-1/2} u), \qquad \|H\| \to 0, \qquad n|H|^{1/2} \to \infty.$$

The NW estimator at $x \in D$ is

$$\widehat{f}_{\mathrm{NW}}(x) = \frac{\mathbf{1}^\top W \boldsymbol{Y}}{\mathbf{1}^\top W \mathbf{1}},$$

where $W := \mathrm{Diag}\big(K_H(X_i - x)\big)$, $\mathbf{1} := (1, \dots, 1)^\top \in \mathbb{R}^n$ and $\boldsymbol{Y} := (Y_1, \dots, Y_n) \in \mathbb{R}^{n \times d_y}$.

To make the analysis easier, we assume the following smoothness requirements.

**Assumption A.1.** *The domain $D$ has $C^2$ boundary (defined as $\partial D$) with principal curvatures uniformly bounded by $\kappa_2$.*

**Assumption A.2.** *The density function $p$ of $X$ satisfies $p \in C^1(D)$. For all dimensions $j \in \{1, \dots, d_y\}$, the function $f$ satisfies $f_j \in C^2(D)$, and the variance function $\sigma^2$ satisfies $\sigma_j^2 \in C(D)$.*

We also assume that the kernel $K$ has easy-to-handle support.

**Assumption A.3.** *$K$ is radial, compactly supported in the unit ball $\mathbb{B}^d := \{u \in \mathbb{R}^d : \|u\| \le 1\} =: \mathrm{supp}(K)$.*

Throughout, $\mathbb{E}[\cdot]$ denotes expectation with respect to the randomness of the training sample $\mathcal{S}_n := \{(X_i, Y_i)\}_{i=1}^n$, while $\int_D (\cdot) \, dx$ denotes the Lebesgue integral over the spatial domain $D$. Because

$$\mathbb{E} \int_D \|\widehat{f}(x) - f(x)\|^2 \, dx = \sum_{j=1}^{d_y} \mathbb{E} \int_D \big(\widehat{f}_j(x) - f_j(x)\big)^2 \, dx,$$

the output dimension $d_y$ only induces a summation across components and does not affect the order; hence, without loss of generality, we take $d_y = 1$ below.

### A.1 INTEGRAL ERROR ESTIMATION OF GLOBAL LINEAR REGRESSION

We consider the global affine class

$$\mathcal{G} := \big\{ g_\theta(x) = \beta_0 + \beta^\top x \ : \ \theta = (\beta_0, \beta^\top)^\top \in \mathbb{R}^{1+d} \big\}.$$

**Lemma A.1.** *If $f \notin \mathcal{G}$, there exists a constant $A_D^* > 0$ such that for every $n$,*

$$\mathbb{E}\bigg[ \int_D \big(\widehat{f}_{\mathrm{GL}}(x) - f(x)\big)^2 \, dx \bigg] \ge A_D^\star.$$

*Proof.* Let $L^2(D, dx) := \{h : D \to \mathbb{R} \text{ with } \int_D h(x)^2 \, dx < \infty\}$ endowed with inner product $\langle h_1, h_2 \rangle = \int_D h_1(x) h_2(x) \, dx$. The set $\mathcal{G}$ is a finite-dimensional linear subspace and hence closed in $L^2(D, dx)$. By the Projection Theorem, the $L^2(D, dx)$-orthogonal projection of $f$ onto $\mathcal{G}$ exists and is unique:

$$g^\dagger \in \arg\min_{g \in \mathcal{G}} \|f - g\|_{L^2(D)}^2 = \arg\min_{g \in \mathcal{G}} \int_D \big(f(x) - g(x)\big)^2 \, dx.$$

Set $A_D^\star := \|f - g^\dagger\|_{L^2(D)}^2$. If $f \notin \mathcal{G}$, then $f - g^\dagger \neq 0$ in $L^2(D)$ and thus $A_D^\star > 0$.

Fix an arbitrary realization $\mathcal{S}_n := (X_i, Y_i)_{i=1}^n$. Since $\widehat{f}_{\mathrm{GL}}(\cdot; \mathcal{S}_n) \in \mathcal{G}$, the optimality of $g^\dagger$ yields

$$\int_D \left(\widehat{f}_{\mathrm{GL}}(x; \mathcal{S}_n) - f(x)\right)^2 dx \;\geq\; \inf_{g \in \mathcal{G}} \int_D \left(g(x) - f(x)\right)^2 dx \;=\; A_D^\star.$$

This inequality holds for every sample $\mathcal{S}_n$. Taking expectation over the training data proves the claim. $\qquad\square$

## A.2 POINT-WISE ERROR ESTIMATION OF LOCAL CONSTANT REGRESSION

In this section we will estimate the point-wise mean-squared-error of NW estimator, whose expressions are given by

$$\mathrm{MSE}_{\mathrm{NW}}(x) := \mathbb{E}\left[\left(\widehat{f}_{\mathrm{NW}}(x) - f(x)\right)^2 \mid \{X_i\}_{i=1}^n\right]$$

$$= \underbrace{\left(\mathbb{E}\left[\widehat{f}_{\mathrm{NW}}(x) - f(x) \mid \{X_i\}_{i=1}^n\right]\right)^2}_{\mathrm{Bias}_{\mathrm{NW}}(x)^2} + \underbrace{\mathbb{E}\left[\left(\widehat{f}_{\mathrm{NW}}(x) - \mathbb{E}\left[\widehat{f}_{\mathrm{NW}}(x) \mid \{X_i\}_{i=1}^n\right]\right)^2 \mid \{X_i\}_{i=1}^n\right]}_{\mathrm{Var}_{\mathrm{NW}}(x)}.$$

When estimating at $x \in D$, the kernel $K_H$ maps the translated domain $(D - x)$ into $\mathrm{supp}(K)$ via the scaling $H^{-1/2}$. The mapped set differs depending on whether $x$ lies in the interior of $D$ or near $\partial D$. This geometric difference drives a larger boundary error for NW, which will underlie the performance gap between NW and LL. We formalize the mapped kernel domain and the boundary layer.

**Definition A.1** (Exact kernel domain and boundary layer.). *For any $x \in D$ and bandwidth $H$, the exact kernel domain is*

$$D_{x,H} := H^{-1/2}(D - x) \cap \mathrm{supp}(K) = \{u \in \mathbb{B}^d : x + H^{1/2}u \in D\}.$$

*Define the boundary layer by*

$$\mathcal{B}(H) := \{\, x \in D : \; D_{x,H} \neq \mathrm{supp}(K) \,\}.$$

We use the following kernel-moment shorthands.

**Definition A.2** (Exact kernel moments). *For all $x \in D$ and bandwidth $H$, we define*

$$\mu_0^\star(x, H) := \int_{D_{x,H}} K(u)\, du, \qquad \mu_1^\star(x, H) := \int_{D_{x,H}} u\, K(u)\, du, \qquad \mu_2^\star(x, H) := \int_{D_{x,H}} uu^\top K(u)\, du.$$

*Define normalized moments $\bar{\mu}_r^\star(x, H) := \mu_r^\star(x, H) / \mu_0^\star(x, H)$.*

Then we are ready to estimate the bias and variance using $H$ and $n$.

**Lemma A.2.** *Under Assumption A.2, we have*

$$\mathrm{Bias}_{\mathrm{NW}}(x) = \nabla f(x)^\top H^{1/2} \bar{\mu}_1^\star(x, H) \;+\; O_p(\|H\|), \quad \mathrm{Var}_{\mathrm{NW}}(x) = \Theta_p\left(\frac{1}{n|H|^{1/2}}\right)$$

*uniformly for all $H \in \mathcal{H}_n$, where $\mathcal{H}_n := \{H = h^2 B : h \in [n^{-a}, n^{-b}], B \succ 0, |B| = 1, \kappa(B) \leq \kappa_1\}$ and $0 < b < a < 1$.*

*Proof.* Defining $\boldsymbol{f} := (f(X_1), \ldots f(X_n))^\top$, for any fixed point $x_0 \in D$, we have

$$\mathrm{Bias}_{\mathrm{NW}}(x_0) = (\mathbf{1}^\top W \mathbf{1})^{-1} \mathbf{1}^\top W (\boldsymbol{f} - f(x_0)\mathbf{1}).$$

Specifically,

$$n^{-1}(\mathbf{1}^\top W \mathbf{1}) = n^{-1} \sum_{i=1}^n K_H(X_i - x_0), \quad n^{-1}\mathbf{1}^\top W(\boldsymbol{f} - f(x_0)\mathbf{1}) = n^{-1} \sum_{i=1}^n K_H(X_i - x_0)(f(X_i) - f(x_0)).$$

By Chebyshev's inequality, for any fixed $H$,

$$n^{-1} \sum_{i=1}^{n} K_H(X_i - x_0) = \int_D K_H(x - x_0)p(x)dx + O_p\left(n^{-1}\sqrt{n \int_D K_H^2(x - x_0)p(x)dx}\right).$$

By Theorem 1 in (Fan et al., 1996), this bound holds *uniformly* over $H \in \mathcal{H}_n$ after multiplying the stochastic term by $\sqrt{\log n}$. Hence,

$$n^{-1} \sum_{i=1}^{n} K_H(X_i - x_0) = \int_D K_H(x - x_0)p(x)dx + O_p\left(n^{-1}\sqrt{n \log n \int_D K_H^2(x - x_0)p(x)dx}\right)$$

$$= \int_{D_{x_0,H}} K(u)p(x_0 + H^{1/2}u)du + o_p(1) = p(x_0)\mu_0^*(x, H) + o_p(1)$$

$$(18)$$

uniformly for $H \in \mathcal{H}_n$.

Similarly,

$$n^{-1} \sum_{i=1}^{n} K_H(X_i - x_0)(f(X_i) - f(x_0))$$

$$= \int_D K_H(x - x_0)p(x)(f(x) - f(x_0))dx + O_p\left(n^{-1}\sqrt{n \log n \int_D K_H^2(x - x_0)(f(x) - f(x_0))^2 p(x)dx}\right)$$

$$= \int_{D_{x_0,H}} K(u)p(x_0 + H^{1/2}u)(f(x_0 + H^{1/2}u) - f(x_0))du + o_p(\|H\|)$$

$$= \int_{D_{x_0,H}} K(u)(p(x_0) + \nabla p(x_0)H^{1/2}u + O(\|H\|))(\nabla f(x_0)^\top H^{1/2}u + O(\|H\|))du + o_p(\|H\|)$$

$$= p(x_0)\nabla f(x_0)^\top H^{1/2}\mu_1^*(x_0, H) + \nabla p(x_0)H^{1/2}\mu_2^*(x_0, H)H^{1/2}\nabla f(x_0) + o_p(\|H\|)$$

$$= p(x_0)\nabla f(x_0)^\top H^{1/2}\mu_1^*(x_0, H) + O_p(\|H\|)$$

$$(19)$$

uniformly for all $H \in \mathcal{H}_n$.

Combining Equations 18, 19, we have

$$\text{Bias}_{\text{NW}}(x_0) = \nabla f(x_0)^\top H^{1/2}\bar{\mu}_1^\star(x_0, H) + O_p(\|H\|)$$

uniformly for all $H \in \mathcal{H}_n$.

Then we calculate variance

$$\text{Var}_{\text{NW}}(x_0) = (\mathbf{1}^\top W \mathbf{1})^{-1}(\mathbf{1}^\top \Sigma \mathbf{1})(\mathbf{1}^\top W \mathbf{1})^{-1}$$

where $\Sigma := \text{diag}(K_H^2(X_i - x_0)\sigma^2(X_i))$. Analogously to equation 18,

$$n^{-1}(\boldsymbol{X}^\top \Sigma \boldsymbol{X}) = n^{-1} \sum_{i=1}^{n} K_H^2(X_i - x_0)\sigma^2(X_i)$$

$$= \int_D K_H^2(x - x_0)\sigma^2(x)p(x)dx + O_p\left(n^{-1}\sqrt{n \log n \int_D K_H^4(x - x_0)\sigma^4(x)p(x)dx}\right)$$

$$= |H|^{-1/2}\int_{D_{x_0,H}} K^2(u)\sigma^2(x_0 + H^{1/2}u)p(x_0 + H^{1/2}u)du + O_p\left(\sqrt{\frac{\log n}{n}}|H|^{-3/4}\right) \quad (20)$$

$$= |H|^{-1/2}\int_{D_{x_0,H}} K^2(u)\sigma^2(x_0)p(x_0)du(1 + o_p(1)) + O_p\left(\sqrt{\frac{\log n}{n}}|H|^{-3/4}\right)$$

$$= |H|^{-1/2}\sigma^2(x_0)p(x_0)\int_{D_{x_0,H}} K^2(u)du(1 + o_p(1))$$

uniformly for all $H \in \mathcal{H}_n$.

Combining Equations 18, 20, we have that

$$\text{Var}_{\text{NW}}(x) = \Theta_p\left(\frac{1}{n|H|^{1/2}}\right)$$

holds uniformly for all $H \in \mathcal{H}_n$. $\qquad\square$

### A.3 INTEGRAL ERROR ESTIMATION OF LOCAL CONSTANT REGRESSION

Here we calculate the integral error for NW estimator. From Lemma A.2 we have that for any $x \in D$, the pointwise bias of NW estimator is

$$\text{Bias}_{\text{NW}}(x) = \nabla f(x)^\top H^{1/2} \bar{\mu}_1^\star(x, H) + O_p(\|H\|).$$

By symmetry of the kernel $K$, the first moment $\bar{\mu}_1^\star(x, H) = 0$ if and only if $x \notin \mathcal{B}(H)$. Roughly, $\text{Bias}_{\text{NW}}(x) = O(\|H^{1/2}\|)$ when $x \in \mathcal{B}(H)$ and $\text{Bias}_{\text{NW}}(x) = O(\|H\|)$ when $x \notin \mathcal{B}(H)$. Hence it is important to bound $\int_{\mathcal{B}(H)} \text{MSE}_{\text{NW}}(x)\, dx$ and $\int_{D \setminus \mathcal{B}(H)} \text{MSE}_{\text{NW}}(x)\, dx$ separately. We first define a smooth substitute for the boundary layer $\mathcal{B}(H)$.

**Definition A.3** (Uniform Boundary Layer). *For any bandwidth $H$ and $\alpha \in (0,1)$, the uniform boundary layer of $D$ is*

$$\mathcal{C}(H, \alpha) := \{x = y - te(y) : y \in \partial D, t \in [0, \alpha h_n(y)]\}$$

*where for any $y \in \partial D$, we define $e(y)$ as the inward unit normal at $y$, and define $h_n(y) := \sqrt{e(y)^\top H\, e(y)}$.*

We now show that one can choose a constant $\alpha \in (0,1)$ (independent of $H$) that "sandwiches" $\mathcal{B}(H)$ between $\mathcal{C}(H, \alpha)$ and $\mathcal{C}(H, \alpha^{-1})$.

**Lemma A.3.** *Under Assumptions A.1, A.3, there exists $\alpha \in (0,1)$ such that for all sufficiently small $H$, we have*

$$\mathcal{C}(H, \alpha) \subset \mathcal{B}(H) \subset \mathcal{C}(H, \alpha^{-1}).$$

*Proof.* We prove the claim with $\alpha = \min\left(\frac{1}{2}, \frac{1}{2}\kappa_1^{-1/2}\right)$.

*Step 1: $\mathcal{C}(H, \alpha) \subset \mathcal{B}(H)$.* Let $\phi$ denote the signed distance to $\partial D$ (positive inside $D$). For $x = y - te(y) \in \mathcal{C}(H, \alpha)$ and any $v \in \mathbb{R}^d$, the standard expansion (using the shape operator $S_y$) gives

$$\phi(x + v) = -t + v_n - \tfrac{1}{2} v_T^\top S_y v_T + O(\|v\|^3),$$

where $v_n := v^\top e(y)$ and $v_T := (I - e(y)e(y)^\top)v$. By Assumption A.1, $\|S_y\|$ is uniformly bounded.

Define

$$u := \kappa_1^{-1/2} h_n(y) H^{-1/2} e(y) \in \mathbb{R}^d.$$

Then

$$\|u\|^2 = \kappa_1^{-1} h_n(y)^2 e(y)^\top H^{-1} e(y) = \kappa_1^{-1} \frac{\left(e(y)^\top He(y)\right)\left(e(y)^\top H^{-1}e(y)\right)}{1} \le \kappa_1^{-1} \kappa(H) \le 1,$$

since $\kappa(H) \le \kappa_1$ by assumption. Hence $u \in \text{supp}(K)$ (Assumption A.3). Using the expansion with $v = H^{1/2}u$ and bounded curvature,

$$\phi\left(x + H^{1/2}u\right) = -t + u^\top H^{1/2} e(y) + O(\|H\|^{3/2}) = -t + \kappa_1^{-1/2} h_n(y) + O(\|H\|^{3/2}).$$

Since $t \le \alpha h_n(y)$ and $\alpha \le \frac{1}{2}\kappa_1^{-1/2}$, for sufficiently small $H$ we have $\phi(x + H^{1/2}u) > 0$, i.e., $x + H^{1/2}u \notin D$. Thus there exists $u \in \text{supp}(K)$ with $u \notin D_{x,H}$, i.e., $D_{x,H} \ne \text{supp}(K)$, so $x \in \mathcal{B}(H)$.

*Step 2: $\mathcal{B}(H) \subset \mathcal{C}(H, \alpha^{-1})$.* Let $x = y - te(y) \in \mathcal{B}(H)$. By definition, there exists $u \in \text{supp}(K)$ with $x + H^{1/2}u \notin D$, i.e.,

$$\phi(x + H^{1/2}u) = -t + u^\top H^{1/2} e(y) + O(\|H\|) > 0.$$

Hence

$$t < u^\top H^{1/2} e(y) + O(\|H\|) \le \|u\| \sqrt{e(y)^\top He(y)} + O(\|H\|) \le h_n(y) + O(\|H\|).$$

For sufficiently small $H$, this yields $t < \alpha^{-1} h_n(y)$ (since $\alpha^{-1} \ge 2$), so $x \in \mathcal{C}(H, \alpha^{-1})$. $\qquad\square$

With $\mathcal{C}(H, \alpha^{-1})$ in hand, we bound the integrated variance and squared bias of the NW estimator.

**Lemma A.4.** *Under Assumptions A.1, A.2, and A.3, we have, uniformly for all $H \in \mathcal{H}_n$,*

$$\int_D \operatorname{Bias}_{\mathrm{NW}}^2(x)\,dx = O_p\big(\|H\|^{3/2}\big) \quad \text{and} \quad \int_D \operatorname{Var}_{\mathrm{NW}}(x)\,dx = \Theta\big(n^{-1}|H|^{-1/2}\big).$$

*Proof.*

$$\int_D \operatorname{Bias}_{\mathrm{NW}}^2(x)dx = \int_{\mathcal{B}(H)} \operatorname{Bias}_{\mathrm{NW}}^2(x)dx + \int_{D\setminus\mathcal{B}(H)} \operatorname{Bias}_{\mathrm{NW}}^2(x)dx$$

$$= O_p(\|H\|)\int_{\mathcal{B}(H)} dx + O_p(\|H^2\|)$$

$$\leq O_p(\|H\|)\int_{\mathcal{C}(H,\alpha^{-1})} dx + O_p(\|H^2\|)$$

$$= O_p(\|H\|)\int_{\partial D}\int_0^{\frac{h_n(y)}{\alpha}} dt\,dS(y) + O_p(\|H^2\|)$$

$$= O_p(\|H^{3/2}\|),$$

since $h_n(y) = \sqrt{e(y)^\top H e(y)} = \Theta(\|H\|^{1/2})$ uniformly under bounded condition number.

For the variance, Lemma A.2 gives $\operatorname{Var}_{\mathrm{NW}}(x) = \Theta_p(n^{-1}|H|^{-1/2})$ pointwise (uniformly over $H \in \mathcal{H}_n$). Integrating over $D$ (whose volume is constant) preserves the order:

$$\int_D \operatorname{Var}_{\mathrm{NW}}(x)dx = \int_D \Theta(n^{-1}|H^{1/2}|^{-1})dx = \Theta(n^{-1}|H^{1/2}|^{-1}).$$

$\square$

## A.4   PROOF OF PROPOSITION 2.1

Combining Lemmas A.1, A.4, we obtain the final conclusion.

**Theorem A.1** (Precise statement of Proposition 2.1). *Under Assumptions A.1, A.2, A.3, if the function $f$ to be estimated is not within the the global affine class $\mathcal{G}$,then*

$$\mathbb{E}\int_D \operatorname{MSE}_{\mathrm{GL}}(x)\,dx = \Omega(1) \quad \text{and} \quad \mathbb{E}\int_D \operatorname{MSE}_{\mathrm{NW}}(x)\,dx = O\big(n^{-3/(d+3)}\big),$$

*where $\widehat{f}_{\mathrm{NW}}$ uses the optimal bandwidth $H \in \mathcal{H}_n$.*

*Proof.* The lower bound for $\operatorname{MSE}_{\mathrm{GL}}$ follows directly from Lemma A.1.

For NW, Lemma A.4 and the definition of $\mathcal{H}_n$ imply

$$\int_D \operatorname{Bias}_{\mathrm{NW}}^2(x)\,dx = O_p(h^3), \qquad \int_D \operatorname{Var}_{\mathrm{NW}}(x)\,dx = \Theta\big(n^{-1}h^{-d}\big)$$

uniformly for $h \in [n^{-a}, n^{-b}]$ (since $\|H\| = \Theta(h^2)$ and $|H|^{1/2} = \Theta(h^d)$ under bounded condition number). Hence

$$\int_D \operatorname{MSE}_{\mathrm{NW}}(x)\,dx = O_p\Big(h^3 + \frac{1}{n\,h^d}\Big) = O_p\big(n^{-3/(d+3)}\big),$$

at the minimizer $h = n^{-1/(d+3)} \in [n^{-a}, n^{-b}]$ of $h^3 + (nh^d)^{-1}$. $\square$

## B   APPENDIX: PROOF OF PROPOSITION 2.2

The local linear estimator at $x \in D$ is

$$\widehat{f}_{\mathrm{LL}}(x) = (\boldsymbol{X}^\top W \boldsymbol{X})^{-1}\boldsymbol{X}^\top W \boldsymbol{Y},$$

where

$$\boldsymbol{X} := \begin{bmatrix} 1 & \cdots & 1 \\ X_1 - x_0 & \cdots & X_n - x_0 \end{bmatrix}^\top.$$

Here we inherit Assumptions A.1, A.2, A.3 and continue to set $d_y = 1$ as in Appendix A.

## B.1 POINT-WISE ERROR ESTIMATION

We have already derived the point-wise error of the local constant estimator in Lemma A.4. We now do the same for the local linear estimator.

**Lemma B.1.** *Under Assumption A.2, we have, uniformly for all $H \in \mathcal{H}_n$,*

$$\text{Bias}_{\text{LL}}(x) = O_p(\|H\|), \qquad \text{Var}_{\text{LL}}(x) = O_p\Big(\frac{1}{n|H|^{1/2}}\Big),$$

*where $\mathcal{H}_n := \{H = h^2 B : h \in [n^{-a}, n^{-b}], B \succ 0, |B| = 1, \kappa(B) \leq \kappa_1\}$ with constants $0 < b < a < 1$.*

*Proof.* For any fixed point $x_0 \in D$, we have

$$\text{Bias}_{\text{LL}}(x_0) = e_1(\boldsymbol{X}^\top W \boldsymbol{X})^{-1} \boldsymbol{X}^\top W(\boldsymbol{f} - \boldsymbol{X}(f(x_0), \nabla f(x_0)^\top)^\top)$$
$$= \frac{1}{2} e_1(\boldsymbol{X}^\top W \boldsymbol{X})^{-1} \boldsymbol{X}^\top W(Q + o_p(\text{tr}(H)))$$

where $Q = [(X_1 - x)^\top \mathcal{H}_f(X_1 - x), \ldots, (X_n - x)^\top \mathcal{H}_f(X_n - x)]^\top$.

$$\text{Var}_{\text{LL}}(x_0) = e_1(\boldsymbol{X}^\top W \boldsymbol{X})^{-1}(\boldsymbol{X}^\top \Sigma \boldsymbol{X})(\boldsymbol{X}^\top W \boldsymbol{X})^{-1} e_1^\top$$

where $\Sigma := \text{diag}(K_H^2(X_i - x_0)\sigma^2(X_i))$.

$$n^{-1}(\boldsymbol{X}^\top W \boldsymbol{X}) = \begin{bmatrix} n^{-1}\sum_{i=1}^n K_H(X_i - x_0) & n^{-1}\sum_{i=1}^n K_H(X_i - x_0)(X_i - x_0)^\top \\ n^{-1}\sum_{i=1}^n K_H(X_i - x_0)(X_i - x_0) & n^{-1}\sum_{i=1}^n K_H(X_i - x_0)(X_i - x_0)(X_i - x_0)^\top \end{bmatrix}$$

$$n^{-1}(\boldsymbol{X}^\top \Sigma \boldsymbol{X}) = \begin{bmatrix} n^{-1}\sum_{i=1}^n K_H^2(X_i - x_0)\sigma^2(X_i) & n^{-1}\sum_{i=1}^n K_H^2(X_i - x_0)\sigma^2(X_i)(X_i - x_0)^\top \\ n^{-1}\sum_{i=1}^n K_H^2(X_i - x_0)\sigma^2(X_i)(X_i - x_0) & n^{-1}\sum_{i=1}^n K_H^2(X_i - x_0)\sigma^2(X_i)(X_i - x_0)(X_i - x_0)^\top \end{bmatrix}$$

$$n^{-1}\boldsymbol{X}^\top W(\boldsymbol{f} - \boldsymbol{X}(f(x_0), \nabla f(x_0)^\top)^\top)$$
$$= \begin{bmatrix} n^{-1}\sum_{i=1}^n K_H(X_i - x_0)(f(X_i) - f(x_0) - \nabla f(x_0)^\top(X_i - x_0)) \\ n^{-1}\sum_{i=1}^n K_H(X_i - x_0)(f(X_i) - f(x_0) - \nabla f(x_0)^\top(X_i - x_0))(X_i - x_0) \end{bmatrix}$$

By the same uniform LLN and $\sqrt{\log n}$ arguments used in the proof of Lemma A.2, we have uniformly over $H \in \mathcal{H}_n$:

$$n^{-1}\sum_{i=1}^n K_H^2(X_i - x_0)\sigma^2(X_i) = |H|^{-1/2}\sigma^2(x_0)p(x_0)R_0(x_0, H)(1 + o_p(1))$$

$$n^{-1}\sum_{i=1}^n K_H^2(X_i - x_0)\sigma^2(X_i)(X_i - x_0)^\top$$
$$= \sigma^2(x_0)p(x_0)|H|^{-1/2}R_1(K)H^{1/2} + O_p\left(\left(\sqrt{\frac{\log n}{n}}|H|^{-3/4} + 1\right)H^{1/2}\mathbf{1}\right)$$

$$n^{-1}\sum_{i=1}^n K_H(X_i - x_0) = p(x_0)\mu_0^*(x_0, H) + o_p(1)$$

$$n^{-1}\sum_{i=1}^n K_H^2(X_i - x_0)\sigma^2(X_i)(X_i - x_0)(X_i - x_0)^\top$$
$$= \sigma^2(x_0)p(x_0)|H|^{-1/2}H^{1/2}R_2(K)H^{1/2} + o_p\left(|H|^{-1/2}H\right)$$

$$n^{-1} \sum_{i=1}^{n} K_H(X_i - x_0)(X_i - x_0)^\top = p(x_0)\mu_1^*(x_0, H)^\top H^{1/2} + O_p(H\mathbf{1})$$

$$n^{-1} \sum_{i=1}^{n} K_H(X_i - x_0)(X_i - x_0)(X_i - x_0)^\top = O_p(H)$$

$$n^{-1} \sum_{i=1}^{n} K_H(X_i - x_0)(f(X_i) - f(x_0) - \nabla f(x_0)^\top(X_i - x_0)) = p(x_0)\mu_2(K)\operatorname{tr}(H\mathcal{H}_f(x_0)) + o_p(\operatorname{tr}(H))$$

$$n^{-1} \sum_{i=1}^{n} K_H(X_i - x_0)(f(X_i) - f(x_0) - \nabla f(x_0)^\top(X_i - x_0))(X_i - x_0) = O_p(H^{3/2}\mathbf{1}),$$

where

$$R_0(x_0, H) = \int_{D_{x_0,H}} K^2(u)du, \quad R_1(x_0, H) = \int_{D_{x_0,H}} K^2(u)udu, \quad R_2(x_0, H) = \int_{D_{x_0,H}} K^2(u)uu^\top du.$$

Then we have the expression of every element in all matrices. A standard blockwise inversion therefore yields

$$\operatorname{Bias}_{\mathrm{LL}}(x) = O_p(\|H\|), \quad \operatorname{Var}_{\mathrm{LL}}(x) = O_p(\frac{1}{n|H|^{1/2}}).$$

uniformly over $H \in \mathcal{H}_n$, proving the claim.

$\square$

## B.2 INTEGRAL ERROR ESTIMATION

From Lemma A.2 we have that for any $x \in D$, the pointwise bias of local constant estimator is

$$\operatorname{Bias}_{\mathrm{NW}}(x) = \nabla f(x)^\top H^{1/2}\bar{\mu}_1^\star(x, H) + O_p(\|H\|).$$

By symmetry of the kernel $K$, the first moment satisfies $\bar{\mu}_1^\star(x, H) = 0$ if and only if $x \notin \mathcal{B}(H)$. We will show that if $f$ has a sufficiently large normal gradient on a measurable subset of $\partial D$, then $\int_{\mathcal{B}(H)} \operatorname{Bias}_{\mathrm{NW}}^2(x)dx$ will have a dominant order $\Theta(\|H^{3/2}\|)$.

We begin by lower-bounding $|\nabla f(x)^\top H^{1/2}\bar{\mu}_1^\star(x, H)|$, to do which we first lower-bound $e(y)^\top H^{1/2}\bar{\mu}_1^*(y - te(y), H)$.

**Lemma B.2.** *Under Assumption A.1, A.3, there exists $\alpha \in (0, 1)$, $c_* > 0$ and $C_* > 0$ such that for all sufficiently small $H$ and all $y \in \partial D$, all $t \in [0, \alpha h_n(y)]$,*

$$|e(y)^\top H^{1/2}\bar{\mu}_1^*(y - te(y), H)| \geq c_* h_n(y) \qquad \|\bar{\mu}_1^*(y - te(y), H)\| \leq C_*.$$

*Proof.* Recall $D_{x,H} = \{u \in \mathbb{B}^d : x + H^{1/2}u \in D\}$. For $x = y - te(y)$ with $y \in \partial D$, choose an orthogonal $Q(y)$ such that $Q(y)H^{1/2}e(y) = h_n(y) e_d$. Define the rotated domain

$$\widetilde{D}_{y,t,H} := \{ v \in \mathbb{B}^d : y - te(y) + H^{1/2}Q(y)v \in D \}.$$

Define the corresponding moments on $\widetilde{D}_{y,t,H}$:

$$\mu_{0,v}^\star(y, t, H) := \int_{\widetilde{D}_{y,t,H}} K(u) \, du, \quad \mu_{1,v}^\star(y, t, H) := \int_{\widetilde{D}_{y,t,H}} u \, K(u) \, du, \quad \bar{\mu}_{1,v}^\star(y, t, H) := \frac{\mu_{1,v}^\star(y, t, H)}{\mu_{0,v}^\star(y, t, H)}.$$

Then

$$\mu_{0,v}^\star(y, t, H) = \mu_0^\star(x, H), \quad Q(y)\,\mu_{1,v}^\star(y, t, H) = \mu_1^\star(x, H), \quad Q(y)\,\bar{\mu}_{1,v}^\star(y, t, H) = \bar{\mu}_1^\star(x, H),$$

and hence

$$e(y)^\top H^{1/2}\bar{\mu}_1^\star(x, H) = h_n(y) \, e_d^\top \bar{\mu}_{1,v}^\star(y, t, H). \tag{21}$$

By the standard signed-distance expansion with shape operator $S_y$ (Assumption A.1), for $v \in \mathbb{R}^d$,

$$\phi\big(y - te(y) + H^{1/2}Q(y)v\big) = -t + h_n(y) v_d - \tfrac{1}{2}z_T^\top S_y z_T + O(\|H\|^{3/2}),$$

where $z_T := (I - e(y)e(y)^\top)H^{1/2}Q(y)v$ and $\|S_y\|$ is uniformly bounded. Therefore

$$y - te(y) + H^{1/2}Q(y)v \in D \iff v_d < \frac{t}{h_n(y)} + O(\|H\|^{1/2}),$$

so

$$\widetilde{D}_{y,t,H} = \left\{ v \in \mathbb{B}^d : v_d < \frac{t}{h_n(y)} + O(\|H\|^{1/2}) \right\}.$$

Consequently,

$$
\begin{aligned}
\mu_{0,v}^\star(y, t, H) &= \int_{\widetilde{D}_{y,t,H}} K(v)\, dv \\
&= \int_{\mathbb{R}^{d-1}} dv_{1:d-1} \int_{-\sqrt{1-\sum_{i=1}^{d-1} v_i^2}}^{t/h_n(y)} K(v_{1:d-1}, v_d)\, dv_d \; + \; O(\|H\|^{1/2}) \\
&\geq \int_{\mathbb{R}^{d-1}} dv_{1:d-1} \int_{-\sqrt{1-\sum_{i=1}^{d-1} v_i^2}}^{0} K(v_{1:d-1}, v_d)\, dv_d \; + \; O(\|H\|^{1/2}).
\end{aligned}
$$

Using positivity and boundedness of $K$ on $\mathbb{B}^d$, there exist $0 < C < D < \infty$ such that

$$C < \mu_{0,v}^\star(y, t, H) < D. \tag{22}$$

Similarly,

$$
\begin{aligned}
e_d^\top \mu_{1,v}^\star(y, t, H) &= \int_{\widetilde{D}_{y,t,H}} v_d K(v)\, dv \\
&= \int_{\mathbb{R}^{d-1}} dv_{1:d-1} \int_{-\sqrt{1-\sum_{i=1}^{d-1} v_i^2}}^{t/h_n(y)} v_d\, K(v_{1:d-1}, v_d)\, dv_d \; + \; O(\|H\|^{1/2}) \\
&=: g\left(\frac{t}{h_n(y)}\right) + O(\|H\|^{1/2}).
\end{aligned}
$$

Note $g(0) < 0$ and $g(\tau)$ decreases as $\tau \downarrow 0$. Choose $\alpha = \alpha_1 \in (0, 1)$ with $g(\alpha_1) < 0$. Then for all $t \in [0, \alpha_1 h_n(y)]$,

$$e_d^\top \mu_{1,v}^\star(y, t, H) \leq g(\alpha_1) + O(\|H\|^{1/2}) < 0. \tag{23}$$

Combining equation 22–equation 23, for small $H$,

$$\left| e_d^\top \bar{\mu}_{1,v}^\star(y, t, H) \right| = \frac{|e_d^\top \mu_{1,v}^\star(y, t, H)|}{\mu_{0,v}^\star(y, t, H)} \geq \frac{|g(\alpha_1)|}{2D}.$$

Using equation 21 yields the first bound with $c_* := |g(\alpha_1)|/(2D)$. The second bound follows from boundedness of $K$ and $\operatorname{supp}(K) \subset \mathbb{B}^d$. $\qquad\square$

Since the leading term of $\mathrm{Bias}_{\mathrm{NW}}(x)$ is $\nabla f(x)^\top H^{1/2}\bar{\mu}_1^\star(x, H)$, the lower bound on $|e(y)^\top H^{1/2}\bar{\mu}_1^\star(y - te(y), H)|$ alone does not guarantee order $\Theta(\|H^{1/2}\|)$; we also require a sufficiently large normal derivative of $f$ along $\partial D$.

**Definition B.1** (Extreme boundary gradient class)**.** *For any domain $D$ and constants $m$ and $M$, we define a class of functions $\mathcal{E}(D, m, M)$, where $f \in \mathcal{E}(D, m, M)$ iff there exist a measurable $\Gamma \subset \partial D$ with $S(\Gamma) > 0$ and constants $m$, $M$ such that $|\partial_e f(y)| \geq m$ and $\|\nabla_T f(y)\| < M$ where $\partial_e f(y) = \nabla f(y)^\top e(y)$ and $\nabla_T f(y) = (I - e(y)e(y)^\top)\nabla f(y)$.*

Then we can prove that if the function to be estimated is within $\mathcal{E}(D, m, M)$ where $m$ and $M$ are specifically chosen constants that are independent of $H$, the NW has an integral squared bias with high order.

**Lemma B.3.** *Under Assumptions A.1, A.2, and A.3, if $f \in \mathcal{E}(D, m, M)$ with*

$$c_*^2 \kappa_1^{-1} m^2 - 2c_* \kappa_1^{-1/2} C_* mM \geq C_1 > 0,$$

*then uniformly over $H \in \mathcal{H}_n$,*

$$\int_D \mathrm{Bias}_{\mathrm{NW}}^2(x)\, dx = \Omega_p(\|H\|^{3/2}) \quad \text{and} \quad \int_D \mathrm{Var}_{\mathrm{NW}}(x)\, dx = \Omega_p(n^{-1}|H|^{-1/2}).$$

**Remark.** *It is easy to construct $f$ and $D$ satisfying Assumptions A.1, A.2 and $f \in \mathcal{E}(D, m, M)$. For example, on $D = \{(x_1, x_2) : x_1^2 + x_2^2 \leq 1\}$, $f(x_1, x_2) = \frac{\sqrt{c_1 \kappa_1}}{2c_*}(x_1^2 + x_2^2)$ works with suitable $m, M$.*

*Proof.* We first lower-bound bias. Choosing $\alpha$ as the minimum $\alpha$ given by Lemmas A.3, B.2, we have

$$\int \text{Bias}_{\text{NW}}^2(x)dx \geq \int_{\mathcal{B}(H)} \text{Bias}_{\text{NW}}^2(x)dx \geq \int_{\mathcal{C}(H,\alpha)} \text{Bias}_{\text{NW}}^2(x)dx$$

$$= \int_{\partial D} \int_0^{\alpha h_n(y)} \text{Bias}_{\text{NW}}^2(y - te(y))\det(I - tS_y)\,dt\,dS(y) \tag{24}$$

$$\geq C_2 \int_\Gamma \int_0^{\alpha h_n(y)} \left( (\nabla f(y - te(y))^\top H^{1/2}\bar{\mu}_1^\star(y - te(y), H))^2 + O_p(||H^{3/2}||) \right) dt\,dS(y),$$

where the last inequality is derived from Lemma A.2 and the boundedness of $S_y$ from Assumption A.1.

For all $t \in \alpha h_n(y)$ and all $\eta \in (0, 1)$,

$$\left( \nabla f(y - te(y))^\top H^{1/2}\bar{\mu}_1^\star(y - te(y), H) \right)^2$$

$$\overset{(a)}{=} \left( \partial_e f(y - te(y))e(y)^\top H^{1/2}\bar{\mu}_1^\star(y - te(y), H) + \nabla_T f(y - te(y))^\top H^{1/2}\bar{\mu}_1^\star(y - te(y), H) \right)^2$$

$$\overset{(b)}{\geq} (1 - \eta)\left( \partial_e f(y - te(y))e(y)^\top H^{1/2}\bar{\mu}_1^\star(y - te(y), H) \right)^2 - \eta^{-1}\left( \nabla_T f(y - te(y))^\top H^{1/2}\bar{\mu}_1^\star(y - te(y), H) \right)^2$$

$$\overset{(c)}{\geq} (1 - \eta)(\partial_e f(y - te(y)))^2 h_n^2(y)c_*^2 - \eta^{-1}||\nabla_T f(y - te(y))||^2 C_*^2||H||$$

$$= (1 - \eta)(\partial_e f(y))^2 h_n^2(y)c_*^2 - \eta^{-1}||\nabla_T f(y)||^2 C_*^2||H|| + O(||H^{3/2}||)$$

$$\geq \left( (1 - \eta)m^2 c_*^2 \kappa_1^{-1} - \eta^{-1}M^2 C_*^2 \right)||H|| + O(||H^{3/2}||).$$

Here $(a)$ uses the fact $\nabla f = \partial_e f(y)e(y) + \nabla_T f(y)$, $(b)$ uses the fact $(A + B)^2 \geq (1 - \eta)A^2 - \eta^{-1}B^2$ for all $\eta \in (0, 1)$, $(c)$ uses Lemma B.2.

Setting $\eta = \frac{MC_* \kappa_1^{1/2}}{mc_*}$, we have

$$\left( \nabla f(y - te(y))^\top H^{1/2}\bar{\mu}_1^\star(y - te(y), H) \right)^2$$

$$\geq \left( (1 - \eta)m^2 c_*^2 \kappa_1^{-1} - \eta^{-1}M^2 C_*^2 \right)||H|| + O(||H^{3/2}||)$$

$$= \left( c_*^2 \kappa_1^{-1}m^2 - 2c_* \kappa_1^{-1/2}C_* mM \right)||H|| + O(||H^{3/2}||) \tag{25}$$

$$\geq C_1||H|| + +O(||H^{3/2}||).$$

Combining Equations 24, 25, we have that

$$\int \text{Bias}_{\text{NW}}^2(x)dx \geq \alpha C_2 \left( C_1||H|| + O_p(||H^{3/2}||) \right) \int_\Gamma h_n(y)dS(y) = \Omega_p(||H^{3/2}||).$$

The last equation holds because $h_n(y) \geq \kappa_1^{-1}||H^{1/2}||$. Using Lemma A.2, it is straightforward to show that

$$\int_\Omega \text{Var}_{\text{NW}}(x)dx = \Omega_p(n^{-1}|H^{1/2}|^{-1}).$$

$\square$

We then show that local linear regression enjoys strictly lower bias than NW on all functions.

**Lemma B.4.** *Under Assumptions A.1, A.2, A.3, we have $\int_D \text{Bias}_{\text{LL}}^2(x)dx = O_p(||H^2||)$ and $\int_D \text{Var}_{\text{LL}}(x)dx = O_p(n^{-1}|H^{1/2}|^{-1})$ uniformly hold for all $H \in \mathcal{H}_n$.*

*Proof.* It is straightforward from Lemma B.1. □

Then it is easy to prove the final conclusion.

**Theorem B.1** (Precise statement of Proposition 2.2)**.** *Under Assumptions A.1, A.2, A.3, if the function $f$ to be estimated is within $\mathcal{E}(D, m, M)$ where*

$$c_*^2 \kappa_1^{-1} m^2 - 2c_* \kappa_1^{-1/2} C_* mM \geq C_1 > 0,$$

*we have* $\mathbb{E} \int_D \mathrm{MSE}_{\mathrm{NW}}(x) dx = \Omega(n^{-3/(d+3)})$ *and* $\mathbb{E} \int_D \mathrm{MSE}_{\mathrm{LL}}(x) dx = O(n^{-4/(d+4)})$*, where both $\widehat{f}_{\mathrm{NW}}$ and $\widehat{f}_{\mathrm{LL}}$ are at their optimal bandwidth $H \in \mathcal{H}_n$.*

*Proof.* According to Lemma B.3 and the definition of $\mathcal{H}_n$, we have that we have $\int_D \mathrm{Bias}_{\mathrm{NW}}^2(x) dx = \Omega_p(h^3)$ and $\int_D \mathrm{Var}_{\mathrm{NW}}(x) dx = \Omega_p(n^{-1}h^{-d})$ uniformly hold for all $h \in [n^{-a}, n^{-b}]$. So we have

$$\int_D \mathrm{MSE}_{\mathrm{NW}}(x) dx = \Omega_p(h^3 + \frac{1}{nh^d}) = \Omega_p(n^{-3/(d+3)})$$

even for optimal $H \in \mathcal{H}_n$. The last equality holds because $h = n^{-1/(d+3)} \in [n^{-a}, n^{-b}]$ is the minimizer of $h^3 + (nh^d)^{-1}$.

According to Lemma B.4, we have that

$$\int_D \mathrm{MSE}_{\mathrm{LL}}(x) dx = O_p(h^4 + \frac{1}{nh^d}) = O_p(n^{-4/(d+4)})$$

for the optimal $H \in \mathcal{H}_n$. According to the definition of $\Omega_p$ and $O_p$, it is straightforward to deduce the conclusion. □

## C  APPENDIX: CONJUGATE GRADIENT SOLVER

The Conjugate Gradient (CG) method (Hestenes & Stiefel, 1952) is an iterative algorithm for solving systems of linear equations with symmetric positive-definite matrices. As described in Section 3, we solve the linear systems with matrix $\Sigma_i$ for blocks of queries in parallel:

$$\Sigma_i x_i = y_i, \quad \text{for } i \in \{(r-1)B_r + 1, \ldots, rB_r\} \tag{26}$$

In the FlashLLA forward algorithm 1, CG is applied within each row block, with $Q_r, M_r, m_r, \omega_r$ being the block quantities, $X_r$ being the solution to be computed and $Y_r$ being the right-hand side. We use the simplest initialization $X_r \leftarrow 0$ for CG. Hence the initial residual $r_i$ is set to be $y_i$, i.e., $R^{(0)} \leftarrow Y_r$ in Algorithm 2.

The core computation is the matrix-vector product $\Sigma_i p_i$ for the search vectors $p_i$ computed in the lines 4-10. The result is stored in matrix $\Sigma_P$. This operation has high I/O intensity due to the requirement to stream through the entire $K$ matrix in HBM during each CG iteration. Consequently, controlling the number of iterations is crucial for both the efficiency and convergence. While maximal iteration number $T \leq d$ can be manually set, further considerations are necessary to ensure the numerical stability and performance.

First, the convergence and convergent rate of CG are greatly influenced by its spectral condition. However, the conditioning varies significantly across positions. For example, for early tokens, the matrix $\Sigma_i$ is low-rank and requires relatively large $\lambda$ to maintain positive definiteness. To address this, we make the regularization $\lambda$ learnable and data-dependent:

$$\lambda_i = \mathrm{sigmoid}(W_\lambda x_i). \tag{27}$$

The dimension of the weight $W_\lambda \in \mathbb{R}^{d \times d_\lambda}$ controls the granularity of the regularization. Setting $d_\lambda = d$ enables per-dimensional regularization, though empirically set $d_\lambda = d_h$ suffices, where $d_h$ denotes the head dimension.

Additionally, since the CG solves multiple systems in parallel, different system converge at different iterations. To prevent the numerical issues from affecting early converged system, we employ an active mask that disables iterations for systems whose residual norm fall below the tolerance $\epsilon$.

---

**Algorithm 2** FlashLLA CG Solver

---

**Require:** Variables $K$ in HBM, $X_r, Y_r, Q_r, M_r, m_r, \omega_r$ in SRAM, block sizes $B_c$, regularization $\lambda$, tolerance $\epsilon$, max iterations $T$, bandwidth $h$.
 1: Initialize on-chip: $R^{(0)} \leftarrow Y_r \in \mathbb{R}^{B_r \times d}$, $P^{(0)} \leftarrow Y_r \in \mathbb{R}^{B_r \times d}$.
 2: **for** $t = 1$ to $T$ **do**
 3:   Initialize on-chip: $\Sigma_P^{(0)} \leftarrow 0 \in \mathbb{R}^{B_r \times d}$.
 4:   **for** $c = 1$ to $\lceil n/B_c \rceil$ **do**
 5:     Load $K_c$ from HBM to SRAM.
 6:     Compute $W = \exp(Q_r K_c^\top / h - \texttt{bcast}(m_r))$.
 7:     Compute $\Sigma_P^{(c)} = \Sigma_P^{(c-1)} + (W \odot P^{(t-1)} K_c) K_c$.
 8:   **end for**
 9:   Compute $P_Q = \texttt{brsum}(Y_r \odot Q_r)$ and $P_M = \texttt{brsum}(P^{(t-1)} \odot M_r)$.
10:   Compute $\Sigma_P^{(\texttt{last})} = \Sigma_P^{(\texttt{last})} - P_Q \odot M_r - P_M \odot Q_r + \texttt{bcast}(\omega_r) \odot P_Q \odot Q_r + \lambda P^{(t-1)}$
11:   Compute $n = \texttt{rsum}(R^{(t-1)} \odot R^{(t-1)})$ and check convergence.
12:   Compute $\alpha = n / \texttt{rsum}(P^{(t-1)} \odot \Sigma_P^{(\texttt{last})})$.
13:   Compute $X^{(t)} = X^{(t-1)} + \texttt{bcast}(\alpha) \odot P^{(t-1)}$.
14:   Compute $R^{(t)} = R^{(t-1)} - \texttt{bcast}(\alpha) \odot \Sigma_P^{(\texttt{last})}$.
15:   Compute $\beta = \texttt{rsum}(R^{(t)} \odot R^{(t)}) / n$.
16:   Compute $P^{(t)} = R^{(t)} + \beta \odot P^{(t-1)}$.
17: **end for**
18: **return** $X_r$ as the solution of $\Sigma_i X = Y_r$ for $i$ in the block.

---

## D  APPENDIX: BACKWARD DERIVATION

This section provides the detailed derivation of the backward pass. Defining the following variables:

$$\hat{b}_i = \sum_{j=1}^{i} s_{ij} v_j = \sum_{j=1}^{i} w_{ij} \frac{n_{ij}}{\delta_i} v_j, \quad g_i = \frac{\partial \mathcal{L}}{\partial \hat{b}_i} \in \mathbb{R}^d \tag{28}$$

$$\gamma_{ij} = g_i^\top v_j = \frac{\partial \mathcal{L}}{\partial s_{ij}}, \quad \beta_i = \frac{1}{\delta_i} \sum_{j=1}^{i} \gamma_{ij} s_{ij}, \quad c_{ij} = \frac{\gamma_{ij} w_{ij}}{\delta_i} \tag{29}$$

The gradient of the loss with respect to $v_j$ is given by:

$$\frac{\partial \mathcal{L}}{\partial v_j} = \sum_{i=j}^{n} s_{ij} g_i, \quad \Delta_V = S^\top G \tag{30}$$

The gradient of $q_i$ and $k_j$ is related to $s_{ij}$, which can be broken down into $w_{ij}$ and $z_{ij}$ path in the computation graph. The partial gradients of the loss with respect to $w_{ij}, z_{ij}$ are given by:

$$\frac{\partial \mathcal{L}}{\partial w_{ij}} = \frac{\partial \mathcal{L}}{\partial s_{ij}} \frac{\partial s_{ij}}{\partial w_{ij}} + \frac{\partial \mathcal{L}}{\partial \omega_i} \frac{\partial \omega_i}{\partial w_{ij}} + \frac{\partial \mathcal{L}}{\partial \mu_i}^\top \frac{\partial \mu_i}{\partial w_{ij}} + \text{Tr}\left( \frac{\partial \mathcal{L}}{\partial \Sigma_i}^\top \frac{\partial \Sigma_i}{\partial w_{ij}} \right) \tag{31}$$

$$= \frac{\gamma_{ij} n_{ij}}{\delta_i} - \beta_i + z_{ij}^\top \frac{\partial \mathcal{L}}{\partial \mu_i} + z_{ij}^\top \frac{\partial \mathcal{L}}{\partial \Sigma_i} z_{ij} \tag{32}$$

$$\frac{\partial \mathcal{L}}{\partial z_{ij}} = \frac{\partial \mathcal{L}}{\partial s_{ij}} \frac{\partial s_{ij}}{\partial z_{ij}} + \frac{\partial \mathcal{L}}{\partial \mu_i}^\top \frac{\partial \mu_i}{\partial z_{ij}} + \text{Tr}\left( \frac{\partial \mathcal{L}}{\partial \Sigma_i}^\top \frac{\partial \Sigma_i}{\partial w_{ij}} \right) \tag{33}$$

$$= -c_{ij} \rho_i + w_{ij} \frac{\partial \mathcal{L}}{\partial \mu_i} + 2 w_{ij} \frac{\partial \mathcal{L}}{\partial \Sigma_i} z_{ij} \tag{34}$$

Then the partial gradients of the loss with respect to $k_j$ and $q_i$ are given by:

$$\frac{\partial \mathcal{L}}{\partial k_j} = \sum_{i=j}^{n} \frac{\partial \mathcal{L}}{\partial w_{ij}} \frac{\partial w_{ij}}{\partial k_j} + \frac{\partial \mathcal{L}}{\partial z_{ij}} \frac{\partial z_{ij}}{\partial k_j} = \sum_{i=j}^{n} \frac{\partial \mathcal{L}}{\partial w_{ij}} \frac{w_{ij}}{h} q_i + \frac{\partial \mathcal{L}}{\partial z_{ij}} \tag{35}$$

$$\frac{\partial \mathcal{L}}{\partial q_i} = \sum_{j=1}^{i} \frac{\partial \mathcal{L}}{\partial w_{ij}} \frac{\partial w_{ij}}{\partial q_i} + \frac{\partial \mathcal{L}}{\partial z_{ij}} \frac{\partial z_{ij}}{\partial q_i} = \sum_{j=1}^{i} \frac{\partial \mathcal{L}}{\partial w_{ij}} \frac{w_{ij}}{h} k_j - \frac{\partial \mathcal{L}}{\partial z_{ij}} \tag{36}$$

In order to compute the partial gradients of $\mu_i$ and $\Sigma_i$, denote

$$u_i = \Sigma_i^{-1} \sum_{j=1}^{i} c_{ij} z_{ij} = \Sigma_i^{-1} \left[ \sum_{j=1}^{i} c_{ij} k_j - \left( \sum_{j=1}^{i} c_{ij} \right) q_i \right] \tag{37}$$

which can be computed with existing Conjugate Gradient solver 2. Then

$$\frac{\partial \mathcal{L}}{\partial \mu_i} = \sum_{j=1}^{i} \frac{\partial \mathcal{L}}{\partial n_{ij}} \frac{\partial n_{ij}}{\partial \mu_i} + \frac{\partial \mathcal{L}}{\partial \delta_i} \frac{\partial \delta_i}{\partial \mu_i} = -u_i + 2\beta_i \rho_i \tag{38}$$

$$\frac{\partial \mathcal{L}}{\partial \Sigma_i} = \sum_{j=1}^{i} \frac{\partial \mathcal{L}}{\partial n_{ij}} \frac{\partial n_{ij}}{\partial \Sigma_i} + \frac{\partial \mathcal{L}}{\partial \delta_i} \frac{\partial \delta_i}{\partial \Sigma_i} = -\frac{1}{2} \rho_i \frac{\partial \mathcal{L}}{\partial \mu_i}^{\top} + \frac{1}{2} u_i \rho_i^{\top} \tag{39}$$

We denote the following variables:

$$\Delta_\mu = -U + 2\mathtt{bcast}(\beta) \odot R \tag{40}$$

We can materialize the gradient of $w_{ij}$ for every $i, j$ pair and then perform the reduction.

$$w_{ij} \frac{\partial \mathcal{L}}{\partial w_{ij}} = \gamma_{ij} s_{ij} + w_{ij} z_{ij}^{\top} \frac{\partial \mathcal{L}}{\partial \mu_i} + w_{ij} z_{ij}^{\top} \frac{\partial \mathcal{L}}{\partial \Sigma_i} z_{ij} \tag{41}$$

We omit the $Q, K$ in the $\mathtt{relmm}$ for simplicity, then the gradient of $w_{ij}$ can be computed as:

$$\Delta_W = \Gamma \odot S + W \odot \left( -\mathtt{bcast}(\beta) + \mathtt{relmm}(\Delta_\mu) - \frac{1}{2} \mathtt{relmm}(\Delta_\mu) \odot \mathtt{relmm}(R) \right) \tag{42}$$

$$+ \frac{1}{2} \mathtt{relmm}(U) \odot \mathtt{relmm}(R)) \tag{43}$$

Then we can compute the gradient of $k_j$ and $q_i$ through $w_{ij}$ branch as follows:

$$\Delta_W^K = \frac{1}{h} \Delta_W^{\top} Q, \quad \Delta_W^Q = \frac{1}{h} \Delta_W K \tag{44}$$

For the $z_{ij}$ path, we avoid materializing the third-order tensor by performing the reduction internally,

$$\sum_{i=j}^{n} \frac{\partial \mathcal{L}}{\partial z_{ij}} = \sum_{i=j}^{n} -c_{ij} \rho_i + \sum_{i=j}^{n} w_{ij} \frac{\partial \mathcal{L}}{\partial \mu_i} + 2 \sum_{i=j}^{n} w_{ij} \frac{\partial \mathcal{L}}{\partial \Sigma_i} z_{ij} \tag{45}$$

$$\sum_{j=1}^{i} \frac{\partial \mathcal{L}}{\partial z_{ij}} = \sum_{j=1}^{i} -c_{ij} \rho_i + \sum_{j=1}^{i} w_{ij} \frac{\partial \mathcal{L}}{\partial \mu_i} + 2 \sum_{j=1}^{i} w_{ij} \frac{\partial \mathcal{L}}{\partial \Sigma_i} z_{ij} \tag{46}$$

Then we can compute the gradient of the loss with respect to $z_{ij}$ as follows:

$$\Delta_Z^K = -C^{\top} R + W^{\top} \Delta_\mu - (W \odot \mathtt{relmm}(\Delta_\mu))^{\top} R + (W \odot \mathtt{relmm}(R))^{\top} U \tag{47}$$

$$\Delta_Z^Q = -\mathtt{brsum}(C) \odot R + \mathtt{brsum}(W) \odot \Delta_\mu \tag{48}$$

$$- \mathtt{brsum}(W \odot \mathtt{relmm}(\Delta_\mu)) \odot R + \mathtt{brsum}(W \odot \mathtt{relmm}(R)) \odot U \tag{49}$$

Hence the gradient of the loss with respect to $k_j$ and $q_i$ can be computed as:

$$\Delta_K = \Delta_W^K + \Delta_Z^K, \quad \Delta_Q = \Delta_W^Q + \Delta_Z^Q \tag{50}$$

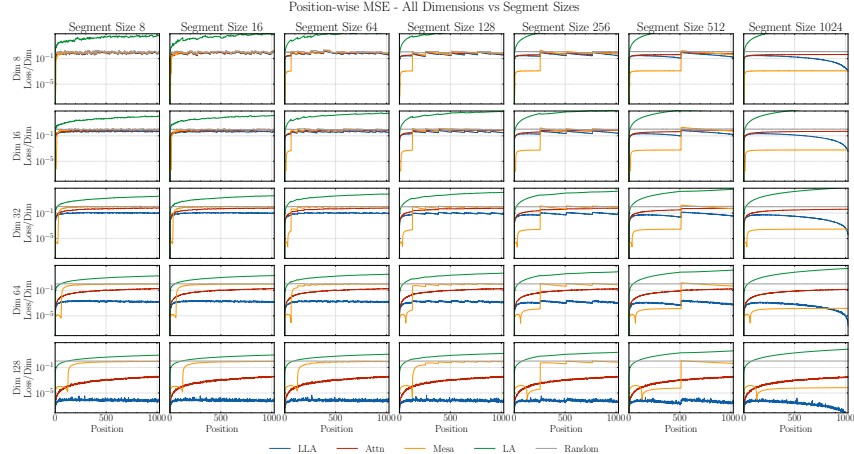

Figure 6: Test-time regression with across all input dimension $d$ and segment size $S$.

## E  APPENDIX: COMPLEXITY ANALYSIS

In this section, we provide a detailed discussion of the computation and IO complexity of FlashLLA and its sparse variants, together with empirical profiling results and numerical precision analysis of our FLASHLLA kernel. This complements the main paper by offering concrete guidance on how computation and memory can be reduced in future work.

### E.1  COMPUTATION AND IO COMPLEXITY

LLA extends the Softmax attention operator by performing regression-style computation across both sequence and feature dimensions. The additional cost arises primarily from solving a $d \times d$ linear system per query. Therefore, the computation of LLA can be amortized over sequential dimensions and feature dimensions. Specifically, we can consider two strategies for reducing the computational and memory cost of LLA:

1. **Sliding-Window LLA (SW-LLA):** Applying a sliding window reduces both computation and IO, particularly for CG-based solvers. Theoretically, local linear regressors has improved sample efficiency compared to local constant methods. Hence, local linear attention has the potential to achieve the same error level with restricted window size.

2. **Block-Diagonal LLA (BD-LLA):** The full LLA formulation captures all feature correlations, which is the dominant source of computational and memory cost and often requires ridge regularization to ensure positive definiteness. Imposing block-diagonal structure reduces both compute and memory while relaxing conditioning requirements. Suppose the block-diagonal structure consists of $g$ diagonal blocks, each of size $d/g$. We consider the following two specifications:

   - $g = d$: only per-feature scaling is modeled. This introduces only $O(nd)$ memory and element-wise multiplications beyond Softmax attention. CG is unnecessary.
   - $g = d/2$: pairwise feature correlations are modeled. This requires inverting $2 \times 2$ matrices per block, which is computationally inexpensive and well-conditioned in practice. CG is unnecessary in this regime.

   These dimensional sparsity strategies often eliminate the need for CG entirely while retaining much of the modeling flexibility of full LLA.

We summarize the per-query complexity of representative methods in Table 1. Here, $n$ denotes the sequence length, $d$ the head dimension, $T$ the number of CG iterations (if used), $g$ the number of diagonal blocks in block-diagonal LLA (BD-LLA), and $B$ the block size used in the kernel.

### E.2  LATENCY PROFILING

| Method | Computation | IO | SW Support |
|---|---|---|---|
| Softmax Attention | $O(nd)$ | $2nd/B$ | ✓ |
| MesaNet | $O(Td^2) + O(d^2)$ | $O(d)$ | ✗ |
| LLA | $O(Td^2) + O(nd)$ | $(T+3)nd/B$ | ✓ |
| BD-LLA (CG) | $O(Td^2/g^2) + O(nd)$ | $(T+3)nd/B$ | ✓ |
| BD-LLA (direct) | $O(d^3/g^2) + O(nd)$ | $3nd/B$ | ✓ |

Table 1: Per-query computation and IO complexity.

We profile latency (ms) and peak memory (GB) during prefill with head dimension 128 and batch size 32. Results are summarized in Figure 7. FLASHLLA reduces working memory and improves latency by $50\times$–$200\times$ over the naive implementation depending on sequence length and CG iterations. Notably, naive LLA runs out-of-memory beyond length 4096, while FlashLLA implementation scales to 8192 tokens with moderate memory growth.

We also evaluate numerical accuracy by comparing our BF16 CG kernel against a naive FP32 implementation that uses `torch.linalg.solve`. Table 2 reports absolute and relative error versus CG iteration count.

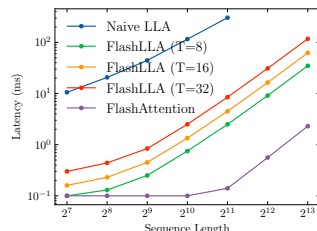

Figure 7: Prefill latency scaling (log-log) with dimension 128 and batch size 32.

Relative error decreases rapidly and reaches an error floor at approximately 16 iteration. Additional iterations yield negligible improvements, indicating that a small constant iteration budget is sufficient in practice.

| $T$ | 1 | 2 | 4 | 8 | 16 | 32 | 64 |
|---|---|---|---|---|---|---|---|
| Absolute Error | 1.49e-3 | 4.51e-4 | 9.13e-5 | 1.98e-5 | 1.72e-5 | 1.72e-5 | 1.72e-5 |
| Relative Error | 1.898 | 0.651 | 0.129 | 0.016 | 0.011 | 0.011 | 0.011 |

Table 2: Numerical precision of BF16 CG kernel versus FP32 solve.

For comparison, if the naive implementation uses BF16: (1) Casting all computations to FP32 yields relative error 0.0173; (2) Casting to FP32 only before computing the final solution yields relative error 0.0174. Our BF16 CG kernel therefore achieves significantly better numerical stability than naive mixed-precision implementations. It also suggests that the CG iterations could be reduced further without sacrificing much accuracy, which would yield additional efficiency gains.

## F   APPENDIX: EXPERIMENTS

### F.1   PIECEWISE LINEAR DATA GENERATION.

Let $n = 2^m$ be the number of segment with $n = \log_2 n \le d$. For each section index $c \in \{1, \ldots, n\}$, define a sign pattern $S_c = (s_{c,1}, s_{c,2}, \ldots, s_{c,m}) \in \{-1, +1\}^m$ by reading the least-significant bits of $c$. For each segment in each sample, draw $Z \sim \mathcal{N}(0, I_d)$ and construct the data $X \in \mathbb{R}^d$ by flipping the first $m$ coordinate of $Z$:

$$X_j = \begin{cases} S_{c,j}|Z_j|, & j \le m \\ Z_j, & j > m \end{cases} \tag{51}$$

As the result, the constructed segment is a truncated Gaussian conditioned to lie in the cone $\mathcal{C}_c = \{x \in \mathbb{R}^d : s_{c,j}x_j \ge 0, j = 1, \ldots, m\}$ where $\mathcal{C}_i \cap \mathcal{C}_j = \emptyset$. Denote $T_c(Z) = (S_c \odot |Z_{1:m}|, |Z_{m+1:d}|)$ and segment distribution $P_c$, we have

$$X \sim P_c \iff X \stackrel{d}{=} T_c(Z) \tag{52}$$

## F.2 Training Configuration

**Test-Time Regression.** This experiment evaluates test-time adaptation without training any model parameters. We sweep over input dimension $d \in \{8, 16, 32, 64, 128\}$ and segment size $S \in \{8, 16, 32, 64, 128, 256, 512, 1024\}$ with fixed sequence length $L = 1024$. Performance is evaluated by averaging the mean squared error over 10,000 independently generated sequences.

**In-Context Regression.** We fix the input and output dimensions at $d_x = d_y = 32$. For each random seed, we generate 100,000 training examples with noise level $\delta = 0.1$ and 1,000 test examples with $\delta = 0$ (noiseless evaluation). For each sequence length $L \in \{64, 128, 256, 512\}$, we sweep over segment size $S \in \{L/8, L/4, L/2, L\}$ and learning rate $\{5 \times 10^{-5}, 10^{-4}, 5 \times 10^{-4}, 10^{-3}\}$. All models are trained with the AdamW optimizer $(\beta_1, \beta_2) = (0.9, 0.999)$, weight decay 0.1, batch size 256, for a maximum of 100 epochs.

**In-Context Associative Recall.** We fix the vocabulary size $|A_k \cup A_v| = 8{,}192$. For each sequence length $L \in \{64, 128, 256, 512\}$, we sweep over the number of key-value pairs $\{L/16, L/8, L/4\}$ and learning rate $\{10^{-4}, 5 \times 10^{-4}, 10^{-3}\}$. We generate 20,000, 40,000, and 60,000 training examples and 1,000 test examples each for the respective key-value pair counts. Training uses AdamW with $(\beta_1, \beta_2) = (0.9, 0.999)$, weight decay 0.1, batch size 256, for a maximum of 32 epochs. Short convolution and feature map are disabled for all models.

**Permutation State Tracking.** We fix the vocabulary size $|A| = 8{,}192$. For each random seed, we generate 100,000 training examples and 1,000 test examples. For each position count $N \in \{16, 24, 48, 96\}$, we sample the number of instructions $S \sim \text{Uniform}(N/6, N/3)$ and use 8 queries per example. We sweep over learning rate $\{10^{-4}, 5 \times 10^{-4}, 10^{-3}\}$. Models are trained with AdamW $(\beta_1, \beta_2) = (0.9, 0.999)$, weight decay 0.1, batch size 256, for a maximum of 64 epochs.

## F.3 Pretraining Analysis

This section supplements the main paper with additional pretraining results discussed in our rebuttal. Our goal is two-fold: (i) report evidence that LLA can yield consistent optimization benefits under settings that recover the kernel-regression perspective, and (ii) characterize qualitative differences in learned attention behavior when training with practical configurations.

**Perplexity with Normalized Queries and Keys.** When queries and keys are normalized, LLA recovers the kernel-regression perspective most directly and empirically shows clear improvement over Softmax Attention in pretraining. In particular, our pretraining curves indicate that smaller ridge parameter $\lambda$ yields better learning curves and lower final perplexity. Concretely, we observe final losses of 3.287 ($\lambda$=0.01), 3.421 ($\lambda$=0.1), 3.509 ($\lambda$=1), and 3.848 for standard Softmax Attention (equivalently, LLA with $\lambda \to \infty$). The results are shown in Figure 8a. While strong QK normalization appears to provide the most stable training in this regime, it can be overly restrictive and may hurt overall performance. Nonetheless, this normalization is commonly adopted by linear-attention architectures (e.g., DeltaNet and MesaNet) as well as some Softmax Attention variants.

**Attention Behavior Analysis.** Under more practical training configurations without strong QK normalization, we observe consistent and interpretable differences in attention behavior that suggest LLA induces a sharper and richer token interaction structure.

To quantify how broadly each token attends, we define an *attention density* metric as the number of positions whose absolute attention score exceeds a threshold (we use $|s_{ij}| > 0.1$). We then average this density across tokens, heads, and layers. We find that Softmax Attention typically concentrates heavily on 1–2 tokens on average, while LLA more often attends to 2–4 tokens across most layers, indicating denser contextual aggregation. The results are shown in Figure 8b.

Although LLA enforces a normalization constraint analogous to attention weights summing to one, it does not constrain the scores to be nonnegative. In practice, LLA exhibits both negative attention scores (as low as $-0.5$) and larger positive scores (up to $\approx 1.8$), whereas Softmax Attention score is restricted to be within $[0, 1]$. Signed contributions allow the model to subtract irrelevant components and amplify useful directions. The results are shown in Figure 8c.

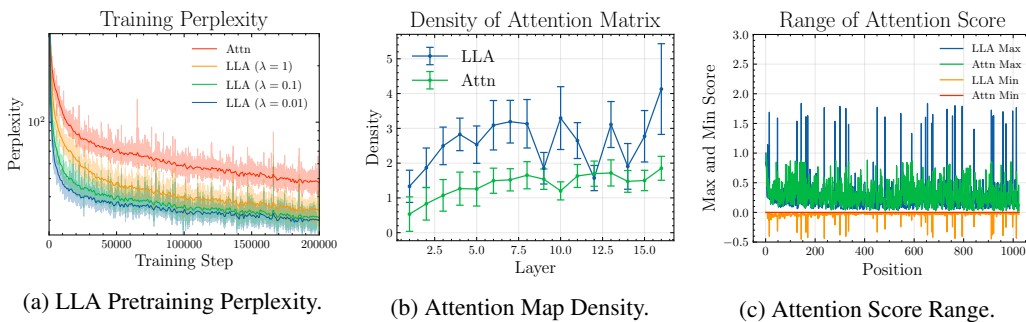

(a) LLA Pretraining Perplexity.  (b) Attention Map Density.  (c) Attention Score Range.

### F.4 ADDITIONAL EXPERIMENT RESULTS

In Figure 6, we provide the full test-time regression results across all input dimension $d$ and segment size $S$. The advantages of LLA scales with the dimension $d$ and nonstationarity. In practical settings, as the dimensionality increases, it is less likely to do exact query $q = k_j$ as in this synthetic experiment. Consequently, kernel selectivity becomes less pronounced when noise is present in query points, limiting the potential advantages of both softmax attention and LLA compared to the ideal conditions of this synthetic experiment. Nevertheless, the overall performance trends remain consistent with our main findings.

Figure 9 shows complete associative recall results across all sequence lengths $L$. For visual clarity, we average results across different numbers of key-value pairs for each sequence length and plot the trajectory of the best score for each model. Gated DeltaNet exhibits distinctive training dynamics characterized by an extended plateau phase with minimal loss improvement, followed by an abrupt transition to significantly lower test loss and corresponding rapid accuracy improvement. The timing of this transition is highly sensitive to hyperparameters such as learning rate and dataset size.

In contrast, LLA demonstrates consistent, gradual improvement in test accuracy with a corresponding smooth decrease in test loss throughout training, mirroring the behavior of Softmax Attention but more powerful. This stable convergence pattern remains robust across a wide range of learning rates and dataset sizes. The marked difference in optimization dynamics suggests fundamental differences in how these models navigate the loss landscape and converge to solutions.

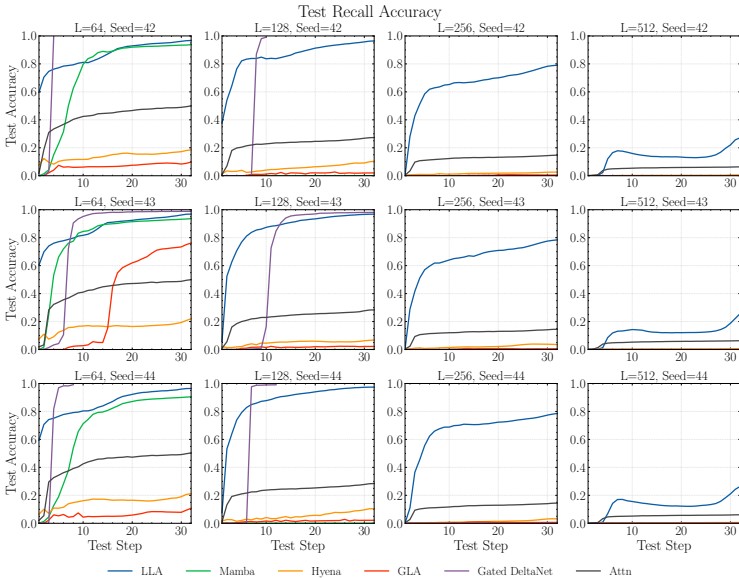

Figure 9: Test accuracy curves for associative recall across all sequence lengths $L$ (averaged over 3 random seeds). Results for different numbers of key-value pairs are averaged within each sequence length for visual clarity

