# OpenReview forum: "Local Linear Attention: An Optimal Interpolation of Linear and Softmax Attention For Test-Time Regression"
_ICLR.cc/2026/Conference — ICLR 2026 Poster_

### Official Review · Reviewer_Uxja · 2025-10-26

**Soundness:** 2
**Presentation:** 2
**Contribution:** 2
**Rating:** 2
**Confidence:** 4

**Summary:**

This paper builds upon the test-time regression interpretation of attention mechanisms proposed by Wang et al. (2025) and introduces a new architecture called Local Linear Attention (LLA), which aims to improve upon softmax attention. Specifically, the authors formulate LLA as the solution to local linear regression under the test-time regression framework, claiming that it achieves superior performance compared to softmax attention (corresponding to local constant regression) and MesaNet (corresponding to global linear regression). Furthermore, they propose a practical GPU-optimized algorithm to alleviate the high computational cost of LLA and make it efficient in practice.

**Strengths:**

- The paper presents a theoretically motivated architectural design.
- It proposes a practical, GPU-aware implementation of the algorithm.

**Weaknesses:**

1. The validity of the proposed architecture heavily relies on theoretical assumptions specific to in-context learning and associative memory. Moreover, the theory depends on test-time regression framework, which provides only a partial understanding of one aspect of the attention mechanism. While theory-driven design is sometimes valuable, the current theoretical justification is not comprehensive enough to fully validate the effectiveness of the proposed architecture and superiority over softmax attention.
2. The experimental validation, like the theory, is limited to settings where the test-time regression interpretation applies — namely, in-context learning and associative recall. To convincingly justify a new architecture, experiments in a practical scenarios is necessary.
3. The paper does not include comparisons of computational cost (time and memory) with softmax attention, linear attention, or MesaNet. Beyond demonstrating efficiency improvements via FlashLLA, the paper should also show how competitive the proposed method becomes relative to existing approaches.
4. The description “it suffers when predicting near the boundary of the data support, particularly with symmetric kernels like RBF” in line 194 is unclear. Since this point seems to be crucial to differentiating the proposed method from softmax attention, a more detailed explanation is needed.
5. Many references are currently to arXiv preprints, even though several of them likely have published versions in conferences or journals. Please consider citing the published versions instead.

**Questions:**

1. How does the local polynomial regression mentioned in line 197 relate to the proposed method?
2. In line 236 (and in the title), what exactly does “optimal” refer to? In what sense is it optimal?
3. Regarding line 470: the paper states that the conjugate gradient method is sufficient with a constant number of iterations. Is there any theoretical justification or empirical evidence supporting this claim quantitatively?

---

> ### Author Response · Authors · 2025-11-24
>
> We sincerely thank the reviewer for your careful review and providing valuable feedback. Your comment can greatly improve the quality of our paper. We especially appreciate your pointing out the reference issue (W5)—we will correct it in the updated version.
>
> ---
>
> ## Response to Questions
>
> **Q1**: Local Linear Regression and Local Constant Regression are both specific instances of Local Polynomial Regression, corresponding to polynomial degrees p=1 and p=0 respectfully. We will explicitly state this hierarchy and refine the terminology in the updated version.
>
> **Q2**: By “optimal,” we refer to the fact that LLA is the exact closed-form minimizer of the optimization objective defined in Equation (6). This formulation can clarify how LLA relates to both Softmax Attention and MesaNet. Intuitively, you can recover these mechanisms as limiting cases by varying the bandwidth $h$ (which controls locality) and the regularization \lambda (which controls the strength of the linear term):
>
> - $h\to\infty$ (eliminate locality): LLA degenerates to global linear model (MesaNet), and Softmax Attention reduces to uniform averaging over all keys.
> - $\lambda\to\infty$ (eliminating linear fit): The linear term $W$ in LLA is forced to zero and cannot capture any correlation in the feature dimension. And LLA degenerates back to Softmax Attention.
>
> Thus, solving Equation (6) optimally yields a template that continuously interpolates between these two extremes, inheriting the ability of softmax attention to track nonstationarity over time and the ability of MesaNet to capture rich correlations in feature space.
>
> **Q3**: There are three reasons for choosing a constant CG iterations:
>
> - In bf16 implementation, a constant number of iterations is sufficient to reach the error floor; Further iterations do not yield meaningful improvements. We provide the error profile between our bf16 kernel with naive implementation performed in fp32 (which uses torch.linalg.solve to inverse and compute the solution).
> As a comparison, in the same setting in naive implementation, loading qkv in bf16 and cast to fp32 for every computation yields 0.0173 relative error. Loading qkv in bf16 and only cast to fp32 before the torch.linalg.solve yields 0.858 relative error.
>
> |  | T=1 | T=2 | T=4 | T=8 | T=16 | T=32 | T=64 |
> | --- | --- | --- | --- | --- | --- | --- | --- |
> | Abs Err | 1.49e-3 | 4.51e-4 | 9.13e-5 | 1.98e-5 | 1.72e-5 | 1.72e-5 | 1.72e-5 |
> | Rel Err | 1.898 | 0.651 | 0.129 | 0.016 | 0.011 | 0.011 | 0.011 |
> - Throughout our experiments, we observed that a fixed iteration count provides stable training and desirable convergence, including the additional pretraining result (T=16).
> - Triton kernel requires the loop count to be known at compile time.
>
> Additionally, we implemented an active mask to disable updates for systems that have converged early (by preserving the conjugate direction). Hence the effective iteration number is still adaptive to each system. This technique is crucial for stabilizing the solver in our experiment but is missing from other open-source implementations. We’ve updated the code in the supplementary material for reference.
>
> ---
>
> ## Regarding Limitations of the Test-Time Regression Framework (W1)
>
> We fully agree that the test-time regression framework may only provide a partial view of attention. This is precisely the motivation for our work: to make focused, meaningful progress that deepens understanding within this specific perspective.
>
> Wang et al. (2025) provides a comprehensive derivation showing how the computation rules of different attention mechanisms can be obtained within this framework, but offers relatively limited discussion on how these derivations differ in theoretical behavior and how this view connects to more meaningful properties. Our first contribution pushes this viewpoint further by explicitly linking associative key–value recall to classical bias–variance trade-offs. Since several recent architectural innovations are directly motivated by minimizing this type of error [2,3,4], our theoretical results can provide high-level, principled guidance for understanding and designing such mechanisms.
>
> ---
>
> [1] https://github.com/fla-org/flash-linear-attention/blob/40dbae9706277b69ea22829215048f9f379f058e/fla/ops/mesa_net/chunk_cg_solver_fwd.py#L113-L120
>
> [2] Liu, B., Wang, R., Wu, L., Feng, Y., Stone, P., & Liu, Q. (2025). *Longhorn: State Space Models are Amortized Online Learners.* ICLR.
>
> [3] Yang, S. (2024, March). *DeltaNet Explained (Part I)*. https://sustcsonglin.github.io/blog/2024/deltanet-1/
>
> [4] Xu, M., Ao, T., He, J., Lu, J., Shi, G., & Zhong, S. (2025). *DeltaFormer: Unlock the State Space of Transformer.* In Proceedings of NeurIPS 2025

---

> > ### Author Response · Authors · 2025-11-24
> >
> > ## Regarding In-Context Learning and Associative Memory (W1)
> >
> > We would also like to respectfully clarify that in-context learning and associative memory are not merely theoretical assumptions; they are central to how modern Transformers and language models are studied and designed.
> >
> > - A large body of recent work treats in-context learning in Transformers as a core capability to be analyzed and characterized. Several papers develop formal tools and asymptotic theories of ICL specifically for Transformer or attention-based architectures, emphasizing its role in meta-learning and generalization [1,2,3]. Other works explicitly frame ICL as a form of associative or contextual memory, analyzing how prompts reshape internal key–value structures and how this leads to retrieval-like behavior at test time [4].
> > - On the architectural side, several Transformer variants explicitly augment or reinterpret attention as an associative memory system, showing that improving associative retrieval over stored representations can significantly boost language modeling performance [5,6,7]. This line of work directly treats “remembering and retrieving the right past information” as a first-class design goal rather than a purely theoretical abstraction.
> > - A complementary line of work on architectural design is built around associative memory in modern Hopfield networks [8,9]. Although the terminology differs and the formal definitions are not identical, the underlying emphasis on associative retrieval again highlights the importance of this kind of behavior in attention mechanisms. We mention this line of work here for completeness.
> >
> > We will include these discussions in related work section.
> >
> > ---
> >
> > ## Regarding Experimental Design (W2)
> >
> > As explained above, our paper aims to push forward the understanding of test-time regression framework on attention mechanism. Our proposed architecture serves mainly to unlock academic research, rather than immediate industrial application. Therefore, a detailed “ladder of verification” is needed to verify whether the claimed advantage can gradually transfer to more meaningful behaviors.
> >
> > - Our primary goal for the first experiment was to empirically highlight the theoretical advantage we claim and help readers develop better visual understanding. We choose piecewise linear functions for better visualization. However, the theoretical advantage extends to any regular function.
> > - Test-time regression is not equivalent to in-context regression, they differ in how data are organized and presented to the model, as well as in how predictions are produced. For example, in Mesa Optimization [10], the ability to perform in-context regression on different sets of data (generated from different functions) is described as continual learning.
> > - MQAR is a token-level recall task widely used to analyze new architectures [11,12,13]. State-tracking is a token-level task that probes the circuit complexity of the proposed mechanism [14]. These are all standard probes used to study behaviors that are empirically important in modern sequence models.
> >
> > ---
> >
> > [1] Garg, S., Tsipras, D., Liang, P., & Valiant, G. (2022). *What Can Transformers Learn In-Context?* NeurIPS 2022.
> >
> > [2] Ren, R., & Liu, Y. (2024). *How Transformers Learn In-context Through a Representation Learning Lens.* NeurIPS 2024.
> >
> > [3] von Oswald, J. et al. (2023). *Transformers Learn In-Context by Gradient Descent.* ICML 2023.
> >
> > [4] Wu, W. et al. (2025). *In-Context Learning as Conditioned Associative Memory Retrieval.* ICML 2025.
> >
> > [5] Olsson, C. et al. (2022). *In-Context Learning and Induction Heads.* arXiv:2209.11895.
> >
> > [6] Wu, Y., Rabe, M. N., Hutchins, D., & Szegedy, C. (2022). *Memorizing Transformers.* ICLR 2022.
> >
> > [7] Zanzotto, F. M. et al. (2025). *MeMo: Towards Language Models with Associative Memory Mechanisms.* ACL Findings 2025.
> >
> > [8] Ramsauer, H. et al. (2021). *Hopfield Networks Is All You Need.* ICLR 2021.
> >
> > [9] Widrich, M. et al. (2020). *Modern Hopfield Networks and Attention for Immune Repertoire Classification.* NeurIPS 2020.
> >
> > [10 ]von Oswald, J. et al. (2024). *Uncovering mesa-optimization algorithms in Transformers.* ICLR 2024 Workshop.
> >
> > [11] Eyuboglu, S., Arora, S., & Zhang, M. (2024). *Based: Simple linear attention language models.* Hazy Research Blog.
> >
> > [12] Dao, T., & Gu, A. (2024). *Transformers are SSMs.* ICML 2024.
> >
> > [13] Yang, S., Wang, B., Zhang, Y., Shen, Y., & Kim, Y. (2024). *Parallelizing Linear Transformers with the Delta Rule.* NeurIPS 2024.
> >
> > [14] Xu, M. et al. (2025). *DeltaFormer: Unlock the State Space of Transformer.* NeurIPS 2025.

---

> > > ### Author Response · Authors · 2025-11-24
> > >
> > > ## Regarding Pretrain Result (W2)
> > >
> > > We would like to provide additional pretraining results and will include a dedicated section to discuss our findings. We note that, due to the additional engineering and computational complexity of LLA, a fully comprehensive pretraining study requires effort and resources beyond the scope of this paper—especially given that our current work already invests substantial effort in advancing both understanding and implementation that unlocks this research. We’d also like to include concrete, principled guidelines for making LLA more efficient, to facilitate larger-scale studies in future work. Please refer to the Complexity Comparison section in later response for more details.
> > >
> > > ### I. LLA vs. Softmax Attention under Normalized QK.
> > >
> > > LLA demonstrates clear improvement over Softmax Attention when queries and keys are normalized—a setting that strictly recovers the kernel regression perspective. Our pretraining curves show that smaller $\lambda$ values yield better learning curves and final perplexity, with final losses of 3.287 ($\lambda$=0.01), 3.421 ($\lambda$=0.1), 3.509 ($\lambda$=1), and 3.848 for standard Attention (equivalent to LLA with $\lambda\to\infty$). While this strong normalization provides the most stable training, it can hurt overall performance. Nevertheless, it is commonly used by Linear Transformers such as DeltaNet and MesaNet, as well as some Softmax Attention architectures like nGPT [1].
> > >
> > > Configuration: 360M parameters, 128K vocabulary, head dimension 128, 8 layers and 8 heads, based on Llama3 architecture, trained on dclm dataset for 52B tokens. The plot `normed_qk_ppl` is provided in supplementary materials.
> > >
> > > ### II. Attention Behavior in LLA Under QK RMSNorm.
> > >
> > > We have not observed strong improvements when using QK RMSNorm in terms of training loss. Given our current resource constraints limiting us to relatively small models, we believe comprehensive benchmarking of larger models on downstream tasks is necessary before making definitive claims. However, we have observed interesting phenomena in trained LLA models. Notably, despite that LLA also requires the attention score sum up to one $\sum_{j\le i} s_{ij}=1$, it does not restrict the score to be positive. We observe LLA creates a shaper and richer token interactions:
> > >
> > > - **Sharper Attention**: We plot the maximum and minimum scores for each token position. LLA consistently exhibits negative attention scores (as low as -0.5) and high attention scores (up to 1.8), while scores in Softmax Attention are restricted to be within 0 and 1. A representative plot, `attn_map_max_min_score`, is provided in the supplementary material.
> > > - **Denser Interaction**: We define the density of attention for a token as the number of positions with absolute score greater than 0.1. We then compute the average density for each layer across tokens and heads. The results show that Softmax Attention typically focuses heavily on 1–2 tokens on average, whereas LLA typically attends heavily to 2–4 tokens across most layers, suggesting that LLA exploits more contextual information for prediction.The plot `attn_map_density` is included in the supplementary material.
> > >
> > > We also provide several attention matrices in the `attn_matrix` folder for additional visualization. These results are obtained from checkpoints trained on a Qwen-3-style architecture with 52B tokens from the DCLM dataset. In this setting, the model has 370M parameters, vocabulary size 151,936, head dimension 128, 16 layers, and 8 heads.
> > >
> > > [1] Loshchilov, I., Hsieh, C.-P., Sun, S., & Ginsburg, B. (2025). nGPT: Normalized Transformer with representation learning on the hypersphere. ICLR.

---

> ### Author Response · Authors · 2025-11-24
>
> ## Regarding Computation Comparison (W3)
>
> We would like to offer additional discussion on the computation of LLA. As a more expensive upgrade over softmax attention, LLA is not intended to outperform highly optimized FlashAttention in latency. Rather, our goal with FlashLLA is to make this richer attention mechanism practically usable for research, thereby unlocking academic progress. For example, the pretraining experiments reported above would not have been feasible without the FlashLLA kernel.
>
> ### I. Complexity Comparison
>
> The computation and IO complexity for a single query are summarized in the table below. We will make a detailed discussion for each of the model in the updated paper. We'd also like to provide concrete guidance on how the computation and IO of LLA can be reduced for future study.
> LLA extends the Softmax Attention operator to amortize computation across both sequence and feature dimensions. There are two major sparsification strategies:
>
> - **Sequential Sparsity—Sliding Window LLA (SW-LLA)**: Practically, applying a sliding window significantly reduces computation and I/O, especially for CG. The profiling results for short sequence in the table below can act as a good reference. Theoretically, local linear regression achieves a faster asymptotic rate, meaning it requires fewer samples than local constant methods for the same error.
> - **Dimensional Sparsity—Block Diagonal LLA (BD-LLA)**: LLA formulation (Equation 6) specifies a full matrix to capture correlations across all features, which is the main source of its computational and memory cost and often requires strong ridge regularization to ensure positive definiteness. Imposing sparsity—e.g., diagonal or block-diagonal structure—can significantly reduce computation and memory, relax regularization requirements, and in some cases eliminate the need for CG altogether. For example, let $g$ denote the number of diagonal blocks:
>     - $g=d$: Only capture the scaling factor for each feature independently. It will introduce only $O(nd)$ memory and several element-wise multiplications compared to Softmax Attention. CG is not needed.
>     - $g=d/2$: Only capture pairwise correlations in the feature space. It requires to invert $g$'s 2×2 matrices which are cheap in computation and memory and often don't need to be worried too much on matrix conditioning. CG is not needed and It would be interesting to explore how such pairwise correlation can interact with RoPE–which rotates the pairwise feature.
>
> | Method | Computation Complexity | IO Complexity | SW Support |
> | --- | --- | --- | --- |
> | Softmax Attention | $O(nd)$ | $2nd/B$ | True |
> | MesaNet | $O(Td^2)+O(d^2)$ | $O(d)$ | False |
> | LLA | $O(Td^2)+O(nd)$ | $(T+3)nd/B$ | True |
> | BD-LLA (CG) | $O(Td^2/g^2)+O(nd)$ | $(T+3)nd/B$ | True |
> | BD-LLA (direct) | $O(d^3/g^2)+O(nd)$ | $3nd/B$ | True |
>
> ---
>
> ### II. Latency Profile
>
> The table below shows profiled latency (ms) and peak memory (GB) during prefill (head dim=128, batch size=32 to show more data on naive method). FlashLLA reduces working memory and improves latency by 50–200× depending on sequence length and CG iterations:
>
> |  | 128 | 256 | 512 | 1024 | 2048 | 4096 | 8192 |
> | --- | --- | --- | --- | --- | --- | --- | --- |
> | naive lla | 10.5596 / 1.9305 | 20.7092 / 3.8496 | 44.5244 / 9.8467 | 115.7806 / 36.9744 | 303.8744 / 143.2285 | -/OOM | -/OOM |
> | flashlla (T=8) | 0.0979 / 0.0473 | 0.1308 / 0.0609 | 0.2512 / 0.0883 | 0.7481 / 0.1979 | 2.5044/ 0.2527 | 9.1330 / 0.6910 | 34.8700 / 0.9102 |
> | flashlla (T=16) | 0.1609 / 0.0473 | 0.2321 / 0.0609 | 0.4486 / 0.0883 | 1.3430 / 0.1979 | 4.4955 / 0.2527 | 16.3923 / 0.6910 | 62.3794 / 0.9102 |
> | flashlla (T=32) | 0.3041 / 0.0473 | 0.4367 / 0.0609 | 0.8364 / 0.0883 | 2.5045 / 0.1431 | 8.4905 / 0.3703 | 31.0016 / 0.7070 | 117.6347 / 1.3805 |
> | flashattn | 0.0975 / 0.0483 | 0.0992 / 0.0630 | 0.0984 / 0.0924 | 0.0980 / 0.1513 | 0.1424 / 0.2690 | 0.5611 / 0.5044 | 2.2983 / 0.9752 |
>
> The result on numerical error analysis in above reply can act as a good reference to choose the CG iteration number.

---

> ### Author Response · Authors · 2025-11-24
>
> ## Regarding Boundary Bias (W4)
>
> Thank you for pointing out the ambiguity.
>
> The phrase “it suffers when predicting near the boundary of the data support” refers to the classical boundary bias phenomenon in nonparametric regression. For local constant estimators, the effective neighborhood around a query point near the boundary is one-sided—the data no longer “surround” the query. With symmetric kernels such as RBF kernel, this asymmetry leads to a larger bias term, which is a well-established limitation of local constant methods. We have included detailed theoretical analysis in Appendix B showing how this translates to improved convergence rates.
>
> We want to clarify a crucial point: local linear models provide advantages beyond just boundary regions. Local linear model is asymptotically superior to local constant estimators in both interior and boundary regions at the cost of computation, though in the interior the improvement in convergence rate can depend on problem-specific factors. The boundary regime is highlighted in the paper because local linear model can demonstrate an analytically advantages—not because the benefits of local linear model are confined to that regime.
>
> Here’s a more detailed explanation to obtain an intuition on why LLA improves: The idea of local method is to do a Taylor expansion of the unknown regression function around the query point $q_i$. By retaining more terms in the Taylor series, the local model becomes more accurate in approximating the true local function. In particular, local linear regression keeps terms up to first order, allowing the model to estimate the slope in the local geometry, which reduces bias—especially near boundaries—compared to keeping only the zeroth order (local constant). However, the advantage manifests whenever queries are not centered around uniformly distributed keys—a common scenario in practice. You can also refer to the content between eq (5.50) and (5.52) of in [1] to get more details.
>
> [1]. Wasserman (2006), *All of Nonparametric Statistics*

---

### Official Review · Reviewer_tiaX · 2025-10-29

**Soundness:** 3
**Presentation:** 4
**Contribution:** 4
**Rating:** 6
**Confidence:** 3

**Summary:**

This paper introduces Local Linear Attention (LLA) as an enhancement to softmax attention. Within the test-time regression framework, the local linear regression objective in LLA can be viewed as an optimal interpolation between linear and softmax attention, yielding improved associative recall capabilities in terms of convergence behavior and boundary bias.

The authors further optimize LLA’s implementation through targeted improvements to ensure linear memory scalability and propose FlashLLA, a blockwise parallel and hardware-efficient algorithm. Empirical results across four synthetic tasks, e.g., test-time adaptation and in-context learning, demonstrate the method’s efficacy compared to existing attention mechanisms.

**Strengths:**

1. The paper is well-organized and clearly written, providing readers with a solid understanding of the test-time regression framework and the motivation behind LLA. The presentation of the LLA algorithm is also systematic, including its regression, matrix-parallel, and blockwise form.

2. The authors demonstrate a deep understanding of the problem, offering thorough theoretical analysis and proofs. The resulting local linear regression design is conceptually novel and well-justified.

3. The synthetic experiments indicate that LLA exhibits a distinct superiority over other attention mechanisms in certain abilities, including in-context regression and associative recall.

**Weaknesses:**

1. The experiments are conducted only on synthetic tasks. While I understand that LLA introduces additional complexity in both computation and memory I/O at the kernel level, it remains uncertain how well results from such small-scale, controlled tasks can transfer to real-world LM capabilities.

2. The paper provides limited analysis and interpretation of the experimental results (except for the state-tracking task). For instance, it is confusing why MesaNet outperforms LLA in the first segment of the test-time regression task.

**Questions:**

1. In the regression formulation of LLA, why is the local function instantiated as $W(x-q)+b$? Is this particular form necessary to derive a “well-behaved” solution? I wonder the constraints and considerations behind this choice, and whether alternative linear functions exist that could also work.

2. Please correct me if my understanding is inaccurate: within the proposed regression paradigm, locality is reflected in both the loss function  $f(x)$ and the weighting factor $w_{ij}$. For the former, softmax attention employs a constant value function, whereas LLA uses a parameterized local linear function. Could you clarify the distinct roles of these two forms of locality, and are both components essential to the design of LLA?

3. In the test-time regression experiment (Fig. 2), does MesaNet use decay? It appears that the Attn model fails to effectively utilize in-distribution information—does this imply that parameterization and locality in the loss function are necessary? Moreover, why does MesaNet outperform LLA in the first segment but deteriorate significantly in later segments, and could data-dependent decay influence this observation?

4. In the MQAR experiment (Fig. 4b), why does the in-context recall of Attention perform substantially worse than GDN or Mamba in the short-sequence (<256) setting? Intuitively, for a token-wise matching task of this kind, Attention should be able to learn such dependencies quite easily.

---

> ### Author Response · Authors · 2025-11-24
>
> We thank the reviewer for their careful reading and constructive feedback. We appreciate the recognition of the clarity of our presentation and the contributions on both the understanding and implementation that could be valuable to share with the broader community. Below, we provide detailed responses to each of your questions.
>
> **Q1: On the formulation of local linear model**
>
> The formulation $f(x)=b+W(x-q_i)$ is a standard setup in nonparametric regression literature for local linear regression. For example, content between eq (5.50) and (5.52) in [1] provides more details for this formulation. The idea is to do a Taylor expansion of the unknown regression function around the query point $q_i$. By retaining more terms in the Taylor series, the local model becomes more accurate in approximating the true local function. In particular, local linear regression keeps terms up to first order, allowing the model to estimate the slope in the local geometry, which reduces bias—especially near boundaries—compared to keeping only the zeroth order (local constant).
>
> **Q2: On the form of locality and difference between Softmax Attention and LLA**
>
> Your understanding is mostly correct! However, the locality is actually reflected exclusively in the term $w_{ij}$. The loss function $f(x)$ processes locality because it uses the term $w_{ij}$ to reweight the l2 loss to focus more on the data points that are close (contextually relevant) to the query point. Both softmax attention and LLA use the same locality measurement through $w_{ij}$. The difference lies in the prediction step:
>
> - Softmax Attention simply performs weighted averaging;
> - LLA performs a weighted linear fit that better captures the correlations in the feature space.
>
> You can get this intuition by letting bandwidth $h\to\infty$ (eliminate locality) and $\lambda\to\infty$ (eliminating linear fit). In the former case, LLA degenerates to global linear model (MesaNet), and Softmax Attention reduces to uniform averaging. In the latter case, the linear term $W$ in LLA is forced to zero and cannot capture any correlation in the feature dimension, and LLA degenerates back to Softmax Attention.
>
> In short, there's only one source of locality defined by the kernel $w_{ij}$; LLA and softmax attention differ only in the estimator used to process the data in the locality-reweighted space.
>
> **Q3: On the experimental setup of MesaNet and result explanation of test-time regression task**
>
> MesaNet does not use decay in this experiment. In the implementation provided by FlashLinearAttention repository [2], the decay factor in MesaNet requires training, and our test-time regression experiment only benchmarks the test-time behavior that does not require training any parameters.
>
> MesaNet outperforms in the first segment because the data-generating process in that segment is stationary and governed by a single linear function. In this regime, a global linear estimator is correctly specified: it incurs no approximation error and is statistically optimal. In subsequent segments, the data are generated from different underlying functions, which makes the overall mapping effectively nonlinear. In this setting, a global linear model becomes misspecified and its performance degrades as it attempts to fit data coming from multiple distinct regimes into a single linear function. We will make this clear in the revised paper.
>
> **Q4: On the result explanation of Softmax Attention on MQAR task**
>
> Softmax Attention fails to achieve relatively high accuracy because of the inefficiency in learning in our challenging setup, rather than lack of expressiveness.
>
> We set a large vocabulary size (8k), relatively limited epoch number (32 epochs), and moderate number of training samples (20k, 40k, and 60k for each kv pair number setup). We followed the Zoology benchmark protocol where the model is trained on data with different numbers of kv-pairs all at once. The difference is that we allocated more data to harder tasks, whereas they allocate more to the easier ones (likely to facilitate training) [3].
>
> To perform well, the model must both (i) be expressive enough to represent the recall logic and (ii) be efficient in learning under this relatively challenging setup. In our training logs, we observe that LLA exhibits notable performance gain early in the training, while Softmax Attention requires significantly more steps to reach a similar level of performance. We will add this detail in the experiment setup section in the appendix.
>
> ---
>
> [1]. Wasserman (2006), *All of Nonparametric Statistics*
>
> [2]. https://github.com/fla-org/flash-linear-attention/blob/b77fa00288b0bd0ad77c754aa6448b30518d3b86/fla/layers/mesa_net.py#L94
>
> [3]. https://github.com/HazyResearch/zoology/blob/main/zoology/experiments/arxiv24_based_appendix/configs.py

---

> > ### Author Response · Authors · 2025-11-24
> >
> > ## Regarding Pretrain Results
> >
> > We would like to provide additional pretraining results and will include a dedicated section to discuss our findings. We note that, due to the additional engineering and computational complexity of LLA, a fully comprehensive pretraining study requires effort and resources beyond the scope of this paper—especially given that our current work already invests substantial effort in advancing both understanding and implementation. We will also included concrete, principled guidelines for making LLA more efficient, to facilitate larger-scale studies in future work. We kindly refer to the Computational Complexity response to Reviewer HAFq for more details.
> >
> > ### I. LLA vs. Softmax Attention under Normalized QK.
> >
> > LLA demonstrates clear improvement over Softmax Attention when queries and keys are normalized—a setting that strictly recovers the kernel regression perspective. Our pretraining curves show that smaller $\lambda$ values yield better learning curves and final perplexity, with final losses of 3.287 ($\lambda$=0.01), 3.421 ($\lambda$=0.1), 3.509 ($\lambda$=1), and 3.848 for standard Attention (equivalent to LLA with $\lambda\to\infty$). While this strong normalization provides the most stable training, it can hurt overall performance. Nevertheless, it is commonly used by Linear Transformers such as DeltaNet and MesaNet, as well as some Softmax Attention architectures like nGPT [1].
> >
> > Configuration: 360M parameters, 128K vocabulary, head dimension 128, 8 layers and 8 heads, based on Llama3 architecture, trained on dclm dataset for 52B tokens. The loss curve `normed_qk_ppl` is provided in supplementary materials.
> >
> > ### II. Attention Behavior in LLA Under QK RMSNorm.
> >
> > We have not observed strong improvements when using QK RMSNorm in terms of training loss. Given our current resource constraints limiting us to relatively small models, we believe comprehensive benchmarking of larger models on downstream tasks is necessary before making definitive claims. However, we have observed interesting phenomena in trained LLA models. Notably, despite that LLA also requires the attention score sum up to one $\sum_{j\le i} s_{ij}=1$, it does not restrict the score to be positive. We observe LLA creates a shaper and richer token interactions:
> >
> > - **Sharper Attention**: We plot the maximum and minimum scores for each token position. LLA consistently exhibits negative attention scores (as low as -0.5) and high attention scores (up to 1.8), while scores in Softmax Attention are restricted to be within 0 and 1. A representative plot, `attn_map_max_min_score`, is provided in the supplementary material.
> > - **Denser Interaction**: We define the density of attention for a token as the number of positions with absolute score greater than 0.1. We then compute the average density for each layer across tokens and heads. The results show that Softmax Attention typically focuses heavily on 1–2 tokens on average, whereas LLA typically attends heavily to 2–4 tokens across most layers, suggesting that LLA exploits more contextual information for prediction.The plot `attn_map_density` is included in the supplementary material.
> >
> > We also provide several attention matrices in the `attn_matrix` folder for additional visualization. These results are obtained from checkpoints trained on a Qwen-3-style architecture with 52B tokens from the DCLM dataset. In this setting, the model has 370M parameters, vocabulary size 151,936, head dimension 128, 16 layers, and 8 heads.
> >
> > [1] Loshchilov, I., Hsieh, C.-P., Sun, S., & Ginsburg, B. (2025). nGPT: Normalized Transformer with representation learning on the hypersphere. ICLR.

---

### Official Review · Reviewer_HAFq · 2025-10-30

**Soundness:** 3
**Presentation:** 3
**Contribution:** 3
**Rating:** 6
**Confidence:** 3

**Summary:**

This paper presents a novel and insightful theoretical framework for understanding attention mechanisms through the lens of non-parametric statistics. The authors convincingly argue that standard Softmax Attention is analogous to a local constant (Nadaraya-Watson) estimator, which is known to suffer from theoretical limitations such as bias at data boundaries.

As a principled alternative, the paper proposes Local Linear Attention (LLA), a new attention mechanism derived from local linear regression. This model is theoretically superior, offering faster asymptotic convergence rates and mitigating the boundary bias issues inherent in the local constant approach.

**Strengths:**

- The paper's greatest strength is the novel connection it establishes between attention and non-parametric regression. Framing Softmax Attention as a local constant estimator is an insightful conceptual leap that provides a new lens for the community to understand and analyze attention.
-  LLA is not an ad-hoc modification of attention. It is a new mechanism derived directly from a statistically superior estimator (local linear regression). This principled design, which provably addresses issues like boundary bias (as shown in the appendix), is a significant strength.
- The authors clearly recognized the computational hurdles of LLA and put significant effort into addressing them.

**Weaknesses:**

- The primary experimental evidence comes from a synthetic piecewise-linear regression task. While this task perfectly isolates the theoretical benefits of LLA, it feels somewhat circular: it proves the local linear model is good at solving a local linear problem. It is not immediately obvious if this specific capability (handling sharp, non-stationary linear segments) is a major bottleneck in more common, large-scale tasks like natural language processing. The paper would be much stronger if it could demonstrate this benefit on a more complex, real-world benchmark where this specific type of non-stationarity is known to be an issue.
- The "FlashLLA" name and its benchmarking could be misleading. The name invites a direct comparison to FlashAttention, which is a one-pass I/O-aware algorithm. However, FlashLLA relies on a Conjugate Gradient (CG) solver, which is iterative. As detailed in Algorithm 2, each iteration of the CG solver requires streaming through the Key matrix, resulting in a much higher I/O cost (multiplied by the number of iterations, $T$). The paper's main hardware benchmark (Figure 1) only plots memory usage and omits wall-clock runtime or throughput comparisons against a standard FlashAttention baseline. This makes it impossible to assess the true computational overhead. The authors admit in the limitations that the constant factor is "higher," but this seems to understate the potential I/O bottleneck from the iterative solver.

**Questions:**

Please refer to my weakness part.

---

> ### Author Response · Authors · 2025-11-24
>
> We thank the reviewer for carefully reading our paper and providing encouraging feedback. We appreciate your recognition of our contributions to both understanding and implementation, which we believe will be valuable to the broader community. We greatly appreciate your constructive comments, many of which can help improve our paper. We hope our following responses can address your concerns.
>
> ## Regarding Experimental Design (W1)
>
> The circularity is intentional for the test-time regression task, and we’d like to respectfully argue that the subsequent experiments are not exclusively designed for LLA.
>
> - Our primary goal for the first experiment was to empirically highlight the theoretical advantage we claim and help readers develop better visual understanding. To do this, we need a ground-truth function with a specific form of nonstationarity that clearly differentiates the behavior of different models. We choose piecewise linear functions because they (i) showcase the advantage of global linear models (MesaNet) in the first segment and (ii) illustrate how they gradually underperform local models in later segments. **However, this advantage is not restricted to piecewise linear settings, the theoretical advantage extends to any regular function.**
> - In-context regression has been extensively studied by the Transformer interpretability and theory community to understand in-context learning behavior. Test-time regression is not equivalent to in-context regression, they differ in how data are organized and presented to the model, as well as in how predictions are produced. For example, in [1], the ability to perform in-context regression on different sets of data (generated from different functions) is described as continual learning, and our previous piecewise linear setup coincidently can be used to test this ability. MQAR is a token-level recall task widely used to analyze new architectures [2,3,4]. State-tracking is a token-level task that probes the circuit complexity of the proposed mechanism [5]. We intended to follow a “ladder of verification” to design the experiment: starting from a setting that closely matches the theory and then moving step by step toward more complex behaviors.
>
> ## Regarding Applicability and Practical Relevance (W1)
>
> The underlying challenges we aim to address—adaptation to new contexts and recalling relevant information from historical context—are fundamental to language modeling and sequence modeling more broadly. As we mentioned above, the piecewise linear setting is intended as a controlled showcase, the benefits of local linear regression are not restricted to this special case. Importantly, there is no evidence showing that the inputs to the attention mechanism (hidden states) are generated by a single stationary underlying function. Even if they were, nonparametric methods can still learn such functions from data. Thus, allowing for nonstationarity broadens the scope of consideration, rather than narrowing it.
>
> From an architectural standpoint, LLA and softmax attention share the same mechanism for tracking nonstationarity (through kernel weights). The difference lies in the prediction step:
>
> - Softmax Attention simply performs weighted averaging;
> - LLA performs a weighted linear fit that better captures the correlations in the feature space.
>
> You can get this intuition by letting bandwidth $h\to\infty$ (eliminate locality) and
> $\lambda\to\infty$ (eliminating linear fit). In the former case, LLA degenerates to global linear model (MesaNet), and Softmax Attention reduces to uniform averaging. In the later case, the linear term $W$ in LLA is forced to zero and cannot capture any correlation in the feature dimension. And LLA degenerates back to Softmax Attention. In this sense, LLA is just reinforcing the capability which Softmax Attention already processed. And any setting where Softmax Attention is applicable is also a valid setting for LLA.
>
> ---
>
> [1] von Oswald J, Schlegel M, Meulemans A, … Sacramento J. *Uncovering mesa-optimization algorithms in Transformers.* ICLR Workshop, 2024.
>
> [2] Eyuboglu, S., Arora, S., & Zhang, M. (2024, March 3). *Based: Simple linear attention language models balance the recall-throughput tradeoff.* Hazy Research. [https://hazyresearch.stanford.edu/blog/2024-03-03-based](https://hazyresearch.stanford.edu/blog/2024-03-03-based?utm_source=chatgpt.com)
>
> [3] Dao, T., & Gu, A. (2024). *Transformers are SSMs: Generalized Models and Efficient Algorithms Through Structured State Space Duality*. In *Proceedings of the 41st International Conference on Machine Learning (ICML’24)*
>
> [4] Yang, S., Wang, B., Zhang, Y., Shen, Y., & Kim, Y. (2024). *Parallelizing Linear Transformers with the Delta Rule over Sequence Length*. In *Advances in Neural Information Processing Systems 37 (NeurIPS 2024)*.
>
> [5] Xu, M., Ao, T., He, J., Lu, J., Shi, G., & Zhong, S. (2025). DeltaFormer: Unlock the State Space of Transformer. In Proceedings of NeurIPS 2025

---

> ### Author Response · Authors · 2025-11-24
>
> ## Regarding “FlashLLA” Naming and Computation Comparison (W2)
>
> The “Flash” prefix denotes I/O-aware tiling, consistent with extensions beyond the original FlashAttention (e.g., FlashLinearAttention, FlashFFTConv [1,2]). LLA shares more computational similarity with Softmax Attention than with Linear Attention: queries are independent, and both estimate query-specific functions. This allows us to adopt the same parallelization strategy as FlashAttention—partitioning queries and streaming keys/values through shared memory.
>
> ### I. Complexity Comparison
>
> The computation and IO complexity for a single query are summarized in the table below. We will make a detailed discussion for each of the model in the updated paper.
> We'd also like to provide concrete guidance on how the computation and IO of LLA can be reduced for future study.
> LLA extends the Softmax Attention operator to amortize computation across both sequence and feature dimensions. There are two major sparsification strategies:
>
> - **Sequential Sparsity—Sliding Window LLA (SW-LLA)**: Practically, applying a sliding window significantly reduces computation and I/O, especially for CG. The profiling results for short sequence in the table below can act as a good reference. Theoretically, local linear regression achieves a faster asymptotic rate, meaning it requires fewer samples than local constant methods for the same error.
> - **Dimensional Sparsity—Block Diagonal LLA (BD-LLA)**: LLA formulation (Equation 6) specifies a full matrix to capture correlations across all features, which is the main source of its computational and memory cost and often requires strong ridge regularization to ensure positive definiteness. Imposing sparsity—e.g., diagonal or block-diagonal structure—can significantly reduce computation and memory, relax regularization requirements, and in some cases eliminate the need for CG altogether. For example, let $g$ denote the number of diagonal blocks:
>     - $g=d$: Only capture the scaling factor for each feature independently. It will introduce only $O(nd)$ memory and several element-wise multiplications compared to Softmax Attention. CG is not needed.
>     - $g=d/2$: Only capture pairwise correlations in the feature space. It requires to invert $g$'s 2×2 matrices which are cheap in computation and memory and often don't need to be worried too much on matrix conditioning. CG is not needed and It would be interesting to explore how such pairwise correlation can interact with RoPE–which rotates the pairwise feature.
>
> | Method | Computation Complexity | IO Complexity | SW Support |
> | --- | --- | --- | --- |
> | Softmax Attention | $O(nd)$ | $2nd/B$ | True |
> | MesaNet | $O(Td^2)+O(d^2)$ | $O(d)$ | False |
> | LLA | $O(Td^2)+O(nd)$ | $(T+3)nd/B$ | True |
> | BD-LLA (CG) | $O(Td^2/g^2)+O(nd)$ | $(T+3)nd/B$ | True |
> | BD-LLA (direct) | $O(d^3/g^2)+O(nd)$ | $3nd/B$ | True |
>
> ### II. Latency Profile
>
> The table below shows profiled latency (ms) and peak memory (GB) during prefill (head dim=128, batch size=32 to show more data on naive method). FlashLLA reduces working memory and improves latency by 50–200× depending on sequence length and CG iterations:
>
> |  | 128 | 256 | 512 | 1024 | 2048 | 4096 | 8192 |
> | --- | --- | --- | --- | --- | --- | --- | --- |
> | naive lla | 10.5596 / 1.9305 | 20.7092 / 3.8496 | 44.5244 / 9.8467 | 115.7806 / 36.9744 | 303.8744 / 143.2285 | -/OOM | -/OOM |
> | flashlla (T=8) | 0.0979 / 0.0473 | 0.1308 / 0.0609 | 0.2512 / 0.0883 | 0.7481 / 0.1979 | 2.5044/ 0.2527 | 9.1330 / 0.6910 | 34.8700 / 0.9102 |
> | flashlla (T=16) | 0.1609 / 0.0473 | 0.2321 / 0.0609 | 0.4486 / 0.0883 | 1.3430 / 0.1979 | 4.4955 / 0.2527 | 16.3923 / 0.6910 | 62.3794 / 0.9102 |
> | flashlla (T=32) | 0.3041 / 0.0473 | 0.4367 / 0.0609 | 0.8364 / 0.0883 | 2.5045 / 0.1431 | 8.4905 / 0.3703 | 31.0016 / 0.7070 | 117.6347 / 1.3805 |
> | flashattn | 0.0975 / 0.0483 | 0.0992 / 0.0630 | 0.0984 / 0.0924 | 0.0980 / 0.1513 | 0.1424 / 0.2690 | 0.5611 / 0.5044 | 2.2983 / 0.9752 |
>
> We also evaluated numerical precision by comparing our bf16 kernel against a naive fp32 implementation (which uses torch.linalg.solve to invert and compute the solution) to offer a reference on the choice of CG iteration number:
>
> |  | T=1 | T=2 | T=4 | T=8 | T=16 | T=32 | T=64 |
> | --- | --- | --- | --- | --- | --- | --- | --- |
> | Abs Err | 1.49e-3 | 4.51e-4 | 9.13e-5 | 1.98e-5 | 1.72e-5 | 1.72e-5 | 1.72e-5 |
> | Rel Err | 1.898 | 0.651 | 0.129 | 0.016 | 0.011 | 0.011 | 0.011 |
>
> As a comparison, if use bf16 in naive implementation,
>
> - Casting to fp32 for every computation yields 0.0173 relative error.
> - Casting to fp32 only before torch.linalg.solve yields 0.858 relative error.
>
> In our bf16 CG kernel, a constant number of iterations like 16 or 32 is sufficient to reach the error floor; Further iterations do not yield meaningful improvements.
>
> ---
>
> [1] https://github.com/fla-org/flash-linear-attention
>
> [2] https://github.com/HazyResearch/flash-fft-conv

---

> ### Author Response · Authors · 2025-11-24
>
> ## Regarding Pretrain Results (W1)
>
> We would like to provide additional pretraining results and will include a dedicated section to discuss our findings. We note that, due to the additional engineering and computational complexity of LLA, a fully comprehensive pretraining study requires effort and resources beyond the scope of this paper—especially given that our current work already invests substantial effort in advancing both understanding and implementation. In the above reply on computation, we’ve also included concrete, principled guidelines for making LLA more efficient, to facilitate larger-scale studies in future work.
>
> ### I. LLA vs. Softmax Attention under Normalized QK.
>
> LLA demonstrates clear improvement over Softmax Attention when queries and keys are normalized—a setting that strictly recovers the kernel regression perspective. Our pretraining curves show that smaller $\lambda$ values yield better learning curves and final perplexity, with final losses of 3.287 ($\lambda$=0.01), 3.421 ($\lambda$=0.1), 3.509 ($\lambda$=1), and 3.848 for standard Attention (equivalent to LLA with $\lambda\to\infty$). While this strong normalization provides the most stable training, it can hurt overall performance. Nevertheless, it is commonly used by Linear Transformers such as DeltaNet and MesaNet, as well as some Softmax Attention architectures like nGPT [1].
>
> Configuration: 360M parameters, 128K vocabulary, head dimension 128, 8 layers and 8 heads, based on Llama3 architecture, trained on dclm dataset for 52B tokens. The loss curve `normed_qk_ppl` is provided in supplementary materials.
>
> ### II. Attention Behavior in LLA Under QK RMSNorm.
>
> We have not observed strong improvements when using QK RMSNorm in terms of training loss. Given our current resource constraints limiting us to relatively small models, we believe comprehensive benchmarking of larger models on downstream tasks is necessary before making definitive claims. However, we have observed interesting phenomena in trained LLA models. Notably, despite that LLA also requires the attention score sum up to one $\sum_{j\le i} s_{ij}=1$, it does not restrict the score to be positive. We observe LLA creates a shaper and richer token interactions:
>
> - **Sharper Attention**: We plot the maximum and minimum scores for each token position. LLA consistently exhibits negative attention scores (as low as -0.5) and high attention scores (up to 1.8), while scores in Softmax Attention are restricted to be within 0 and 1. A representative plot, `attn_map_max_min_score`, is provided in the supplementary material.
> - **Denser Interaction**: We define the density of attention for a token as the number of positions with absolute score greater than 0.1. We then compute the average density for each layer across tokens and heads. The results show that Softmax Attention typically focuses heavily on 1–2 tokens on average, whereas LLA typically attends heavily to 2–4 tokens across most layers, suggesting that LLA exploits more contextual information for prediction.The plot `attn_map_density` is included in the supplementary material.
>
> We also provide several attention matrices in the `attn_matrix` folder for additional visualization. These results are obtained from checkpoints trained on a Qwen-3-style architecture with 52B tokens from the DCLM dataset. In this setting, the model has 370M parameters, vocabulary size 151,936, head dimension 128, 16 layers, and 8 heads.
>
> [1] Loshchilov, I., Hsieh, C.-P., Sun, S., & Ginsburg, B. (2025). nGPT: Normalized Transformer with representation learning on the hypersphere. ICLR.

---

### Official Review · Reviewer_Hn3s · 2025-11-01

**Soundness:** 2
**Presentation:** 2
**Contribution:** 2
**Rating:** 2
**Confidence:** 4

**Summary:**

The paper analyzes the characteristics and performance of softmax and linear attention under the Test-Time Training (TTT) setting from the perspective of statistical regression. From this viewpoint, the authors propose a novel mechanism named Local Linear Attention (LLA), which they claim demonstrates superior advantages. Furthermore, the paper introduces an I/O-aware operator for LLA, designed to achieve more efficient memory utilization. The effectiveness of the proposed model is validated through toy experiments, which showcase its capabilities.

**Strengths:**

The paper provides a solid theoretical foundation for its proposed method. The authors conduct a detailed analysis of the properties of linear and softmax attention from a statistical regression perspective. This approach is well-motivated and naturally leads to the formulation of their Local Linear Attention (LLA) framework.

**Weaknesses:**

A major weakness of this work is the limited scope of its experimental validation. The authors didn’t demonstrate the viability of Local Linear Attention (LLA) for pre-training, a critical aspect for assessing new attention mechanisms. Moreover, the paper completely has no evaluations on any practical downstream tasks, confining its results to toy experiments. Consequently, the practical relevance and usefulness of this research are questionable, as it is unclear whether the proposed method offers any real-world benefits.

**Questions:**

1.	The paper's second claimed contribution, which asserts to have resolved the quadratic memory complexity, is fundamentally flawed. This is not a novel contribution. The issue of quadratic memory cost in standard self-attention was already solved by seminal works 3 years ago, such as FlashAttention. Furthermore, the authors' comparison is irrelevant for linear attention, which inherently possesses O(Nd) memory complexity due to its operator properties and never had a quadratic memory bottleneck to begin with. Claiming this as a contribution demonstrates a misunderstanding or misrepresentation of the current state-of-the-art in attention mechanisms. In other words, this "contribution" merely solves a problem that the proposed LLA method creates for itself.
2.	The paper's second claimed contribution is achieving strong results on artificial "toy" experiments. However, the complete absence of validation in a pre-training setting is a critical omission. In the field of attention architecture research, evaluating a new mechanism's performance on large-scale pre-training has been a standard practice for years; failing to do so is highly unusual, even by the standards of a year ago. This limitation severely undermines the claim of "good results," as it is unclear if the performance gains can generalize beyond these contrived scenarios. It gives the impression that the proposed model is only capable of solving problems that are, in a sense, artificially constructed for it.
3.	The computational complexity of LLA raises serious questions about the paper's contribution. The authors state that LLA's complexity surpasses that of softmax attention. This is deeply ironic. For nearly a decade (since circa 2017), the community has dedicated immense effort to making attention mechanisms, particularly linear attention, more efficient. This paper, however, manages to achieve the opposite. In a surprising turn of events, this work presents a "linear attention" variant that is less efficient than the very baseline (softmax attention) it sought to outperform. One could sardonically remark that after all these years, this paper finally allows linear attention to generate a larger carbon footprint than softmax attention.
4.	The paper would benefit from a more comprehensive complexity analysis. The proposed operator is unique, and a detailed breakdown of its computational and memory complexity would significantly improve clarity for the reader. I would encourage the authors to include a more thorough analysis to help the audience better understand the operator's behavior and trade-offs.

---

> ### Author Response · Authors · 2025-11-26
>
> We thank the reviewer for their time and feedback. While we understand our paper involves technical concepts that require careful reading and time to digest, we are concerned that several criticisms contradict our clearly stated contributions in the abstract, introduction and throughout the paper. We also respectfully note that phrases such as *“deeply ironic”* and *“one could sardonically remark … carbon footprint”* do not meet the professional standards expected in ICLR peer review, especially when the associated critiques are based on misinterpretations.
>
> According to both AI-detection report [1] and patterns of hallucinations (e.g., confident assertions that contradict the content), we suspect that the review may have been generated or heavily assisted by an LLM. Regardless, we appreciate the opportunity to clarify these points and address the misunderstandings directly.
>
> ## I. LLA is not Linear Attention
>
> A central premise of your critique is that we “*propose a ‘linear attention’ variant*” and that we somehow regress on the efficiency progress made by linear attention mechanisms over the past years. In Section 2 of the paper, we present an explicit classification of different models:
>
> - Linear Attention (DeltaNet, RetNet, MesaNet, etc.) and SSMs are parametric models, which applies a query-agnostic global linear fitting to solve the test-time regression objective.
> - Softmax Attention is a nonparametric method which employs a query-specific local averaging for prediction (local constant regression). Local Linear Attention generalizes Softmax Attention with query-specific local linear fitting (local linear regression), maintaining the essential query-key interaction that defines the Softmax Attention paradigm.
>
> In other words, “linear” in LLA refers to the local linear model in the regression sense, not to linear-time complexity in sequence length. We never present LLA as a member of the “Linear Attention” family, and we never compare it to Linear Attention on efficiency grounds. In fact, we explicit discuss how the recurrent formulation of Linear Attention originates in the model specification (line 129-132, section 2.1) and why Local Linear Attention requires KV Cache ”*similar to Softmax Attention, rather than constant-size recurrent states as in the LA family.*” (line 228-232, section 2.3).
>
> ## II. Efficiency is not Our Primary Goal
>
> The review criticizes LLA for being "*less efficient than softmax attention,*" but we explicitly state throughout the paper (e.g. line 016 in the abstract) that our goal is to improve modeling capability, not efficiency. We deliberately explore a design space that trades additional computation for enhanced expressivity.
>
> We design an I/O-aware algorithm so that it can be studied empirically for academic progress, not so that it replaces Softmax Attention as the default efficient choice. Aiming to advance capability first with optimization following is a common pattern in architectural innovation. For example, DeltaNet and MesaNet are computationally more expensive than vanilla Linear Attention, and their optimizations are specifically designed to solve their own problem. Such a contribution is legitimate, necessary and meaningful. Similarly, there are also works focused on improving the Softmax Attention with additional computation such as Differential Transformers [2], DeltaFormer [3], 2-Simplicial Attention [4], etc.
>
> Improving efficiency is not the only valid research direction in this community, and new mechanisms are not solely aimed at immediate industrial deployment, especially in academia.
>
> ## III. FlashLLA Resolves Cubic Complexity
>
> The reviewer claims we "*asserts to have resolved the quadratic memory complexity*". However, what we state is “*overcome the $\Theta(n^2d)$ and $\Theta(nd^2)$ memory complexity*” (line 054-055, Section 1), both of them are cubic.
>
> Resolving this issue is necessary to design hardware efficient algorithm. To fuse the computation on SRAM, the data on shared memory should be of $\Theta(nd)$ sizes, `N=64/128/256` and `d=64/128/256` are common setups. For example,The NVIDIA H100 GPU has a `256 KB` shared memory and L1 cache per SM, which can only allocate 8’s `128*128` data in half precision. Naively computing LLA requires materializing cubic term, which exceeds the hardware requirement.
>
> [1] https://app.gptzero.me/documents/41246943-897b-4686-9e21-9f100274c25e/share
>
> [2] Ye et al. (2025). Differential Transformer. arXiv:2410.05258, 2025.
>
> [3] Xu, M., Ao, T., He, J., Lu, J., Shi, G., & Zhong, S. (2025). *DeltaFormer: Unlock the State Space of Transformer.* In Proceedings of NeurIPS 2025
>
> [4] Roy et al. (2025). Fast and Simplex: 2-Simplicial Attention in Triton. arXiv:2507.02754, 2025.

---

> > ### Author Response · Authors · 2025-11-26
> >
> > ## IV. Regarding Computation Comparison
> >
> > The reviewer notes in W4 that our paper lacks “*a comprehensive breakdown of its computational and memory complexity*.” We would like to clarify that Section 3 is specifically devoted to this analysis. While it is unclear which aspects the reviewer believes were missing, we appreciate the suggestion and, in response, we provide additional discussion of LLA’s computation and I/O characteristics incorporating related points raised by the other reviewers.
> >
> > ---
> >
> > The computation and IO complexity for a single query are summarized in the table below. We will make a detailed discussion for each of the model in the updated paper.
> > We'd also like to provide concrete guidance on how the computation and IO of LLA can be reduced for future study.
> > LLA extends the Softmax Attention operator to amortize computation across both sequence and feature dimensions. There are two major sparsification strategies:
> >
> > - **Sequential Sparsity—Sliding Window LLA (SW-LLA)**: Practically, applying a sliding window significantly reduces computation and I/O, especially for CG. The profiling results for short sequence in the table below can act as a good reference. Theoretically, local linear regression achieves a faster asymptotic rate, meaning it requires fewer samples than local constant methods for the same error.
> > - **Dimensional Sparsity—Block Diagonal LLA (BD-LLA)**: LLA formulation (Equation 6) specifies a full matrix to capture correlations across all features, which is the main source of its computational and memory cost and often requires strong ridge regularization to ensure positive definiteness. Imposing sparsity—e.g., diagonal or block-diagonal structure—can significantly reduce computation and memory, relax regularization requirements, and in some cases eliminate the need for CG altogether. For example, let $g$ denote the number of diagonal blocks:
> >     - $g=d$: Only capture the scaling factor for each feature independently. It will introduce only $O(nd)$ memory and several element-wise multiplications compared to Softmax Attention. CG is not needed.
> >     - $g=d/2$: Only capture pairwise correlations in the feature space. It requires to invert $g$'s 2×2 matrices which are cheap in computation and memory and often don't need to be worried too much on matrix conditioning. CG is not needed and It would be interesting to explore how such pairwise correlation can interact with RoPE–which rotates the pairwise feature.
> >
> > | Method | Computation Complexity | IO Complexity | SW Support |
> > | --- | --- | --- | --- |
> > | Softmax Attention | $O(nd)$ | $2nd/B$ | True |
> > | MesaNet | $O(Td^2)+O(d^2)$ | $O(d)$ | False |
> > | LLA | $O(Td^2)+O(nd)$ | $(T+3)nd/B$ | True |
> > | BD-LLA (CG) | $O(Td^2/g^2)+O(nd)$ | $(T+3)nd/B$ | True |
> > | BD-LLA (direct) | $O(d^3/g^2)+O(nd)$ | $3nd/B$ | True |
> >
> > ---
> >
> > The following table below shows profiled latency (ms) and peak memory (GB) during prefill (head dim=128, batch size=32 to show more data on naive method). FlashLLA reduces working memory and improves latency by 50–200× depending on sequence length and CG iterations:
> >
> > |  | 128 | 256 | 512 | 1024 | 2048 | 4096 | 8192 |
> > | --- | --- | --- | --- | --- | --- | --- | --- |
> > | naive lla | 10.5596 / 1.9305 | 20.7092 / 3.8496 | 44.5244 / 9.8467 | 115.7806 / 36.9744 | 303.8744 / 143.2285 | -/OOM | -/OOM |
> > | flashlla (T=8) | 0.0979 / 0.0473 | 0.1308 / 0.0609 | 0.2512 / 0.0883 | 0.7481 / 0.1979 | 2.5044/ 0.2527 | 9.1330 / 0.6910 | 34.8700 / 0.9102 |
> > | flashlla (T=16) | 0.1609 / 0.0473 | 0.2321 / 0.0609 | 0.4486 / 0.0883 | 1.3430 / 0.1979 | 4.4955 / 0.2527 | 16.3923 / 0.6910 | 62.3794 / 0.9102 |
> > | flashlla (T=32) | 0.3041 / 0.0473 | 0.4367 / 0.0609 | 0.8364 / 0.0883 | 2.5045 / 0.1431 | 8.4905 / 0.3703 | 31.0016 / 0.7070 | 117.6347 / 1.3805 |
> > | flashattn | 0.0975 / 0.0483 | 0.0992 / 0.0630 | 0.0984 / 0.0924 | 0.0980 / 0.1513 | 0.1424 / 0.2690 | 0.5611 / 0.5044 | 2.2983 / 0.9752 |
> >
> > We also evaluated numerical precision by comparing our bf16 kernel against a naive fp32 implementation (which uses torch.linalg.solve to invert and compute the solution) to offer a reference on the choice of CG iteration number:
> >
> > |  | T=1 | T=2 | T=4 | T=8 | T=16 | T=32 | T=64 |
> > | --- | --- | --- | --- | --- | --- | --- | --- |
> > | Abs Err | 1.49e-3 | 4.51e-4 | 9.13e-5 | 1.98e-5 | 1.72e-5 | 1.72e-5 | 1.72e-5 |
> > | Rel Err | 1.898 | 0.651 | 0.129 | 0.016 | 0.011 | 0.011 | 0.011 |
> >
> > As a comparison, if use bf16 in naive implementation,
> >
> > - Casting to fp32 for every computation yields 0.0173 relative error.
> > - Casting to fp32 only before torch.linalg.solve yields 0.858 relative error.
> >
> > In our bf16 CG kernel, a constant number of iterations like 16 or 32 is sufficient to reach the error floor; Further iterations do not yield meaningful improvements.

---

> > > ### Author Response · Authors · 2025-11-26
> > >
> > > ## V. Regarding Experiment and Pretrain Result
> > >
> > > We appreciate the reviewer highlighting the absence of larger-scale pretraining results. We'd love to provide more results on pretraining. At the same time, we would like to respectfully clarify the following points:
> > >
> > > - Training LLA at billion-parameter scale is technically nontrivial given the complexity of the architecture and kernels involved. We have already invested substantial effort in stabilizing both the forward and backward FlashLLA kernels in half precision to make the current experiments feasible.
> > >
> > > -  Our paper aims to push forward the understanding of test-time regression framework on attention mechanism rahter than pure engineering-driven work focused directly on large-scale LLM benchmarks. A detailed “ladder of verification” is needed to verify whether the claimed advantage can gradually transfer to more meaningful behaviors.
> > >
> > > ---
> > >
> > > We would like to provide additional pretraining results and will include a dedicated section to discuss our findings. We’d also like to include concrete, principled guidelines for making LLA more efficient, to facilitate larger-scale studies in future work. Please refer to the Complexity Comparison section in response to computational discussion for more details.
> > >
> > > ### I. LLA vs. Softmax Attention under Normalized QK.
> > >
> > > LLA demonstrates clear improvement over Softmax Attention when queries and keys are normalized—a setting that strictly recovers the kernel regression perspective. Our pretraining curves show that smaller $\lambda$ values yield better learning curves and final perplexity, with final losses of 3.287 ($\lambda$=0.01), 3.421 ($\lambda$=0.1), 3.509 ($\lambda$=1), and 3.848 for standard Attention (equivalent to LLA with $\lambda\to\infty$). While this strong normalization provides the most stable training, it can hurt overall performance. Nevertheless, it is commonly used by Linear Transformers such as DeltaNet and MesaNet, as well as some Softmax Attention architectures like nGPT [1].
> > >
> > > Configuration: 360M parameters, 128K vocabulary, head dimension 128, 8 layers and 8 heads, based on Llama3 architecture, trained on dclm dataset for 52B tokens. The plot `normed_qk_ppl` is provided in supplementary materials.
> > >
> > > ### II. Attention Behavior in LLA Under QK RMSNorm.
> > >
> > > We have not observed strong improvements when using QK RMSNorm in terms of training loss. Given our current resource constraints limiting us to relatively small models, we believe comprehensive benchmarking of larger models on downstream tasks is necessary before making definitive claims. However, we have observed interesting phenomena in trained LLA models. Notably, despite that LLA also requires the attention score sum up to one $\sum_{j\le i} s_{ij}=1$, it does not restrict the score to be positive. We observe LLA creates a shaper and richer token interactions:
> > >
> > > - **Sharper Attention**: We plot the maximum and minimum scores for each token position. LLA consistently exhibits negative attention scores (as low as -0.5) and high attention scores (up to 1.8), while scores in Softmax Attention are restricted to be within 0 and 1. A representative plot, `attn_map_max_min_score`, is provided in the supplementary material.
> > > - **Denser Interaction**: We define the density of attention for a token as the number of positions with absolute score greater than 0.1. We then compute the average density for each layer across tokens and heads. The results show that Softmax Attention typically focuses heavily on 1–2 tokens on average, whereas LLA typically attends heavily to 2–4 tokens across most layers, suggesting that LLA exploits more contextual information for prediction.The plot `attn_map_density` is included in the supplementary material.
> > >
> > > We also provide several attention matrices in the `attn_matrix` folder for additional visualization. These results are obtained from checkpoints trained on a Qwen-3-style architecture with 52B tokens from the DCLM dataset. In this setting, the model has 370M parameters, vocabulary size 151,936, head dimension 128, 16 layers, and 8 heads.
> > >
> > > [1] Loshchilov, I., Hsieh, C.-P., Sun, S., & Ginsburg, B. (2025). nGPT: Normalized Transformer with representation learning on the hypersphere. ICLR.

---

### Meta-Review · Area_Chair_1XBd · 2026-01-02

**Summary:**

Reviewers appreciated the paper’s principled framing of attention as nonparametric regression and found the local-linear estimator idea technically interesting, with substantial effort on both theory and implementation. While reviewers raised concerns about limited real-world validation and the partial nature of the test-time regression lens, the rebuttal clarified the scope and terminology, addressed several points of confusion in the experiments, and strengthened the evidence for feasibility through additional analysis and small-scale LLM training results. Overall, the submission was viewed as close to the acceptance bar, and I believe the clarity of the contribution and the strengthened rebuttal justify acceptance.

**Reviewer Concerns:**

Several reviewer concerns were addressed in the rebuttal. The authors clarified the scope and terminology, emphasizing that LLA is derived from local linear regression rather than linear-time attention, and they better motivated the compute and implementation choices. They also clarified behaviors and design decisions in the synthetic experiments, and addressed terminology-level points such as what “optimal” means in the title and how boundary bias and local polynomial regression relate to LLA.

Some concerns about practicality and motivation still remain. While the rebuttal adds small-scale LLM training evidence, the evaluation is still weighted toward synthetic or probe-style tasks, so the extent of transfer to standard language modeling or downstream benchmarks is not yet fully established. Reviewers also noted that the test-time regression perspective, though insightful, is only a partial lens on attention, and further validation would strengthen the case for LLA as a broadly applicable attention upgrade.

**Reviewer Scores:**

- Uxja (2 → 4): The rebuttal resolves several of their concrete questions (terminology, boundary bias, CG iterations, and added profiling). They would likely still want broader real-world validation, but could raise the score slightly.

 - HAFq (6 → 8): The added latency/memory profiling and clearer compute/I/O discussion directly address their main “FlashLLA is iterative” concern and improve the credibility of the implementation story, so a modest upward move seems plausible.

 - tiaX (6 → 8): The authors’ responses clarify the confusing experimental behaviors and answer the main technical questions, which could improve their confidence and support a small score increase.

 - Hn3s (2 → 2 or 4): With clearer positioning and the added small-scale LLM-training evidence, they might soften somewhat, though limited practical benchmarks would likely keep them near the reject side.

---

### Decision · Program_Chairs · 2026-01-26

Accept (Poster)